# On Fitting Flow Models with Large Sinkhorn Couplings

**Stephen Zhang**[*]                                                   *syz@syz.id.au*
*Apple, University of Melbourne*

**Alireza Mousavi-Hosseini**[*]                                   *mousavi@cs.toronto.edu*
*Apple, University of Toronto*

**Michal Klein**                                                     *michalk@apple.com*
*Apple*

**Marco Cuturi**                                                     *cuturi@apple.com*
*Apple*

**Reviewed on OpenReview:** *https://openreview.net/forum?id=3MLKJZgY62*

## Abstract

Flow models transform data gradually from one modality (e.g. noise) onto another (e.g. images). Such models are parameterized by a time-dependent velocity field, trained to fit segments connecting pairs of source and target points. When the pairing between source and target points is given, training flow models boils down to a supervised regression problem. When no such pairing exists, as is the case when generating data from noise, training flows is much harder. A popular approach lies in picking source and target points independently (Lipman et al., 2023). This can, however, lead to velocity fields that are slow to train, but also costly to integrate at inference time. *In theory*, one would greatly benefit from training flow models by sampling pairs from an optimal transport (OT) measure coupling source and target, since this would lead to a highly efficient flow solving the Benamou & Brenier dynamical OT problem. *In practice*, recent works have proposed to sample *mini-batches* of $n$ source and $n$ target points and reorder them using an OT solver to form *better* pairs. These works have advocated using batches of size $n \approx 256$, and considered OT solvers that return couplings that are either sharp (using e.g. the Hungarian algorithm) or blurred (using e.g. entropic regularization, a.k.a. Sinkhorn). We follow in the footsteps of these works by exploring the benefits of increasing this mini-batch size $n$ by three to four orders of magnitude, and look more carefully at the effect of the entropic regularization $\varepsilon$ used in the Sinkhorn algorithm. Our analysis is facilitated by new scale-invariant quantities to report the sharpness of a coupling. Our sharded computations across multiple GPUs and GPU nodes allow scaling up $n$. We show that in both synthetic and image generation tasks, flow models greatly benefit when fitted with large Sinkhorn couplings, with a low entropic regularization $\varepsilon$.

## 1 Introduction

Finding a map that can transform a source into a target measure is a task at the core of generative modeling and unpaired modality translation. Following the widespread popularity of GAN formulations (Goodfellow et al., 2014), the field has greatly benefited from a gradual, time-dependent parameterization of these transformations as normalizing flows (Rezende & Mohamed, 2015) and neural ODEs (Chen et al., 2018). Such flow models are now commonly estimated using flow matching (Lipman et al., 2024). While a velocity formulation substantially increases the expressivity of generative models, this results, on the other hand, in

---

[*]Equal contributions. Work done during an internship at Apple.

a higher cost at inference time due to the additional burden of running an ODE solver. Indeed, a common drawback of Neural-ODE solvers is that they require potentially many steps, and therefore many passes through the flow network, to generate data. In principle, to mitigate this problem, the gold standard for such continuous-time transformations is given by the solution of the Benamou & Brenier dynamical optimal transport (OT) problem, which should be equivalent, if trained perfectly, to a 1-step generation achieved by the Monge map formulation (Santambrogio, 2015, §1.3). In practice, while the mathematics (Villani, 2003) of optimal transport have contributed to the understanding of these methods (Liu et al., 2022), it remains unclear whether tools from the computational OT toolbox (Peyré & Cuturi, 2019), which is typically used to compute large scale couplings on data (Klein et al., 2025), can decisively help with the estimation of flows in high-dimensional / high-sample sizes regimes.

**Stochastic interpolants.** The flow matching (FM) framework (Lipman et al., 2024), introduced in concurrent seminal papers (Peluchetti, 2022; Lipman et al., 2023; Albergo & Vanden-Eijnden, 2023; Neklyudov et al., 2023) proposes to estimate a flow model by leveraging a pre-defined interpolation $\mu_t$ between source $\mu_0$ and target $\mu_1$ measures — the stochastic interpolant following the terminology of Albergo & Vanden-Eijnden. That interpolation is the crucial ingredient used to fit a parameterized velocity field with a regression loss. In practice, such an interpolation can be formed by sampling $X_0 \sim \mu_0$ independently of $X_1 \sim \mu_1$ and defining $\mu_t$ as the law of $X_t := (1-t)X_0 + tX_1$. One can then fit a parameterized time-dependent velocity field $\mathbf{v}_\theta(t, \mathbf{x})$ that minimizes the expectation of $\|X_1 - X_0 - \mathbf{v}_\theta(X_T, T)\|^2$ w.r.t. $X_0, X_1$ and $T$ a random time variable in $[0, 1]$. This procedure (hereafter abbreviated as Independent-FM, I-FM) has been immensely successful, but can suffer from high variance, and as highlighted by Liu (2022) the I-FM loss can never be zero. Furthermore, minimizing it cannot recover an optimal transport path: the effect of this can be measured by noticing a high curvature when integrating the ODE needed to form an output from an input sample point $\mathbf{x}_0$.

**From I-FM to Batch-OT FM.** To fit exactly the OT framework, ideally one would choose $\mu_t$ to be the McCann interpolation between $\mu_0$ and $\mu_1$, which would be $\mu_t := ((1-t)\mathrm{Id} + tT^\star)_\# \mu_0$, where $T^\star$ is the Monge map connecting $\mu_0$ to $\mu_1$. Unfortunately, this insight is irrelevant, since knowing $T^\star$ would mean that no flow needs to be trained at all. Adopting a more practical perspective, Pooladian et al. (2023) and Tong et al. (2023) have proposed to modify I-FM and select pairs of source and target points more carefully, using discrete OT solvers. Concretely, they sample mini-batches $\mathbf{x}_0^1 \ldots, \mathbf{x}_0^n$ from $\mu_0$ and $\mathbf{x}_1^1, \ldots, \mathbf{x}_1^n$ from $\mu_1$; compute an $n \times n$ OT coupling matrix; sample pairs of indices $(i_\ell, j_\ell)$ from that bistochastic matrix, and feed the flow model with pairs $\mathbf{x}_0^{i_\ell}, \mathbf{x}_1^{j_\ell}$. This approach, referred to as Batch OT-FM in the literature, was recently used and adapted in Tian et al. (2024); Generale et al. (2024); Klein et al. (2023); Davtyan et al. (2025); Kim et al. (2024). Despite their appeal, these modifications have not yet been widely adopted. The consensus stated recently by Lipman et al. (2024) seems to be still that *"the most popular class of affine probability paths is instantiated by the independent coupling"*.

**Can mini-batch OT really help?** We try to answer this question by noticing first that the evaluations carried out in all of the references cited above use batch sizes of $2^8 = 256$ points, more rarely $2^{10} = 1024$, upper bounded by $2^{12} = 4096$ for Kim et al. (2024). We believe that for many of these works this might be due to a reliance on the Hungarian algorithm (Kuhn, 1955) whose $O(n^3)$ complexity is prohibitive for large $n$. We also notice that, while these works also consider entropic OT (EOT) (Cuturi, 2013), they stick to a single $\varepsilon$ regularization value in their evaluations (e.g. 0.2 Kim et al. (2024)). We go back to the drawing board in this paper to study whether batch OT-FM can reliably work, and if so, for which regimes of mini-batch size $n$, regularization $\varepsilon$, and data dimension $d$. Our contributions are:

- Rather than drawing an artificial line between Batch-OT (in Hungarian or EOT form) and I-FM, we leverage the fact that *all* of these approaches can be interpolated using EOT: Hungarian corresponds to the case where $\varepsilon \to 0$ while I-FM is recovered with $\varepsilon \to \infty$. I-FM is therefore a particular case of Batch-OT with infinite regularization, which can be continuously moved towards batch-OT.

- We modify the Sinkhorn algorithm when used with the squared-Euclidean cost: we drop norms and only use negative dot-product. This improves stability and still returns the correct solution.

- We define a renormalized entropy for couplings, to pin them efficiently on a scale of 0 (bijective assignment induced by a permutation, e.g. that returned by the Hungarian algorithm) to 1 (independent coupling).

This quantity is useful because, unlike transport cost or entropy regularization $\varepsilon$, it is bounded in $[0, 1]$ and does not depend on the data dimension $d$ or coupling size $n \times n$.

- We explore in our experiments substantially different regimes for $n$ and $\varepsilon$. We vary the mini-batch size from $n = 2^{11} = 2048$ to $n = 2^{21} = 2{,}097{,}152$ and consider an adaptive grid to set $\varepsilon$ that results in Sinkhorn couplings whose normalized entropy is distributed within $[0, 1]$.

## 2 Background Material on Optimal Transport and Flow Matching

Let $\mathcal{P}_2(\mathbb{R}^d)$ denote the space of probability measures over $\mathbb{R}^d$ with finite second moment. Let $\mu, \nu \in \mathcal{P}_2(\mathbb{R}^d)$, and let $\Gamma(\mu, \nu)$ be the set of joint probability measures in $\mathcal{P}_2(\mathbb{R}^d \times \mathbb{R}^d)$ with left-marginal $\mu$ and right-marginal $\nu$. The OT problem in its Kantorovich formulation is:

$$W_2(\mu, \nu)^2 := \inf_{\pi \in \Gamma(\mu, \nu)} \iint \frac{1}{2} \|x - y\|^2 \mathrm{d}\pi(x, y). \tag{1}$$

A minimizer of (1) is called an *OT coupling measure*, denoted $\pi^\star$. If $\mu$ was a noise source and $\nu$ a data target measure, $\pi^\star$ would be the perfect coupling to sample pairs of noise and data to learn flow models: sample $\mathbf{x}_0, \mathbf{x}_1 \sim \pi^\star$ and ensure the flow models bring $\mathbf{x}_0$ to $\mathbf{x}_1$. Such optimal couplings $\pi^\star$ are in fact induced by *pushforward maps*: when paired optimally, a point $\mathbf{x}_0$ can only be associated with a $\mathbf{x}_1 = T(\mathbf{x}_0)$, where $T : \mathbb{R}^d \to \mathbb{R}^d$ is the Monge optimal transport map, defined as follows:

$$T^\star(\mu, \nu) := \underset{T : T_\# \mu = \nu}{\arg\min} \int \frac{1}{2} \|\mathbf{x} - T(\mathbf{x})\|^2 \mathrm{d}\mu(\mathbf{x}) \tag{2}$$

where the push-forward constraint $T_\# \mu = \nu$ means that for $X \sim \mu$ one has $T(X) \sim \nu$. Monge OT maps have been characterized by Brenier in great detail:

**Theorem 1** (Brenier (1991))**.** *If $\mu \in \mathcal{P}_2(\mathbb{R}^d)$ has an absolutely continuous density then (2) is solved by a map $T^\star$ of the form $T^\star = \nabla u$, where $u : \mathbb{R}^d \to \mathbb{R}$ is convex. Moreover if $u$ is a convex potential that is such that $\nabla u_\# \mu = \nu$ then $\nabla u$ solves (2).*

As a result of Theorem 1, one can choose an arbitrary convex potential $u$, a starting measure $\mu$, and define a synthetic task to train flow matching models between $\mu_0 := \mu$ and $\mu_1 := \nabla u_\# \mu$, for which a ground truth coupling $\pi^\star$ is known. Inspired by Korotin et al. (2021) who considered the same result to benchmark Monge (1781) map solvers, we use this setting in § 4.2 to benchmark batch-OT.

**Entropic OT.** Entropic regularization (Cuturi, 2013) has become the most popular approach to estimate a finite sample analog of $\pi^\star$ using samples $(\mathbf{x}_1, \ldots, \mathbf{x}_n)$ and $(\mathbf{y}_1, \ldots, \mathbf{y}_n)$. Using a regularization strength $\varepsilon > 0$, a cost matrix $\mathbf{C} := [\frac{1}{2} \|\mathbf{x}_i - \mathbf{y}_j\|^2]_{ij}$ between these samples, the entropic OT (EOT) problem can be presented in primal and dual forms as:

$$\min_{\mathbf{P} \in \mathbb{R}_+^{n \times n}, \mathbf{P}\mathbf{1}_n = \mathbf{P}^T\mathbf{1}_n = \mathbf{1}_n/n} \langle \mathbf{P}, \mathbf{C} \rangle - \varepsilon H(\mathbf{P}), \quad \max_{\mathbf{f}, \mathbf{g} \in \mathbb{R}^n} \frac{1}{n} \langle \mathbf{f} + \mathbf{g}, \mathbf{1}_n \rangle - \varepsilon \langle \exp\left(\frac{\mathbf{f} \oplus \mathbf{g} - \mathbf{C}}{\varepsilon}\right), \mathbf{1}_{n \times n} \rangle, \tag{3}$$

where $H(\mathbf{P}) = -\langle \mathbf{P}, \log(\mathbf{P}) \rangle$ is the discrete entropy functional and $\mathbf{P}$ is the bistochastic coupling matrix.

The optimal solutions to (3) are usually found with the Sinkhorn algorithm, as presented in Algorithm 1, where for a matrix $\mathbf{S}$ we write $\min_\varepsilon(\mathbf{S}) := [-\varepsilon \log\left(\mathbf{1}^\top e^{-\mathbf{S}_{i\cdot}/\varepsilon}\right)]_i$, and $\oplus$ is the tensor sum of two vectors, i.e. $(\mathbf{f} \oplus \mathbf{g})_{ij} := \mathbf{f}_i + \mathbf{g}_j$. The optimal dual variables (3) $(\mathbf{f}^\varepsilon, \mathbf{g}^\varepsilon)$ can then be used to instantiate a valid coupling matrix $\mathbf{P}^\varepsilon = \exp\left((\mathbf{f}^\varepsilon \oplus \mathbf{g}^\varepsilon - \mathbf{C})/\varepsilon\right)$, which approximately solves the finite-sample counterpart of (1). An important remark is that as $\varepsilon \to 0$, the solution $\mathbf{P}^\varepsilon$ converges to the optimal transport matrix solving (1), while $\mathbf{P}^\varepsilon \to \frac{1}{n^2}\mathbf{1}_{n \times n}$ as $\varepsilon \to \infty$. These two limiting points coincide with the

---

**Algorithm 1** $\mathrm{SINK}(\mathbf{X} \in \mathbb{R}^{n \times d}, \mathbf{Y} \in \mathbb{R}^{n \times d}, \varepsilon, \tau)$

1: $\mathbf{f}, \mathbf{g} \leftarrow \mathbf{0}_n, \mathbf{0}_n$.
2: $\mathbf{C} \leftarrow [\frac{1}{2}\|\mathbf{x}_i - \mathbf{y}_j\|^2]_{ij}, i \leq n, j \leq n$
3: **while** $\| \exp\left(\frac{\mathbf{f} \oplus \mathbf{g} - \mathbf{C}}{\varepsilon}\right)\mathbf{1}_n - \frac{1}{n}\mathbf{1}_n\|_1 > \tau$ **do**
4: $\quad \mathbf{f} \leftarrow \varepsilon \log \frac{1}{n}\mathbf{1}_n + \min_\varepsilon(\mathbf{C} - \mathbf{f} \oplus \mathbf{g}) + \mathbf{f}$
5: $\quad \mathbf{g} \leftarrow \varepsilon \log \frac{1}{n}\mathbf{1}_n + \min_\varepsilon(\mathbf{C}^\top - \mathbf{g} \oplus \mathbf{f}) + \mathbf{g}$
6: **end while**
7: **return** $\mathbf{f}, \mathbf{g}, \mathbf{P} = \exp\left((\mathbf{f} \oplus \mathbf{g} - \mathbf{C})/\varepsilon\right)$

---

*optimal assignment* matrix (or optimal permutation as returned e.g. by the Hungarian algorithm (Kuhn, 1955)), and the uniform independent coupling used implicitly in I-FM.

**Independent and Batch-OT Flow Matching.** FM methods use a stochastic interpolant $\mu_t$ with law $X_t := (1 - t)X_0 + tX_1$, to minimize the expectation of a squared-norm regression loss $\min_\theta \mathbb{E}_{T,X_0,X_1}\|X_1 - X_0 - \mathbf{v}_\theta(X_T,T)\|^2$ where $X_0 \sim \mu_0, X_1 \sim \mu_1$ and $T$ a random variable in $[0,1]$. In I-FM, this interpolant is implemented by taking independent batches of samples $\mathbf{x}_0^1 \ldots, \mathbf{x}_0^n$ from $\mu_0$, $\mathbf{x}_1^1, \ldots, \mathbf{x}_1^n$ from $\mu_1$, and $t_1, \ldots, t_n$ time values sampled in $[0,1]$, to form the loss values $\|\mathbf{x}_1^k - \mathbf{x}_0^k - \mathbf{v}_\theta((1 - t_k)\mathbf{x}_0^k + t_k\mathbf{x}_1^k, t_k)\|^2$. In the formal-

---

**Algorithm 2** FM 1-Step$(\mu_0, \mu_1, n, \text{OT-SOLVE})$

1: $\mathbf{X}_0 = (\mathbf{x}_0^1, \ldots, \mathbf{x}_0^n) \sim \mu_0$
2: $\mathbf{X}_1 = (\mathbf{x}_1^1, \ldots, \mathbf{x}_1^n) \sim \mu_1$
3: $\mathbf{P} \leftarrow \text{OT-SOLVE}(\mathbf{X}_0, \mathbf{X}_1)$ or $\mathbf{I}_n/n$
4: $(i_1, j_1), \ldots, (i_n, j_n) \sim \mathbf{P}$
5: $t_1, \ldots, t_n \leftarrow \text{TIMESAMPLER}$
6: $\tilde{\mathbf{x}}^k \leftarrow (1 - t_k)\mathbf{x}_0^{i_k} + t_k\mathbf{x}_1^{j_k}$, for $k \le n$
7: $\mathcal{L}(\theta) = \sum_k \|\mathbf{x}_1^{j_k} - \mathbf{x}_0^{i_k} - \mathbf{v}_\theta(\tilde{\mathbf{x}}^k, t_k)\|^2$
8: $\theta \leftarrow \text{GRADIENT-UPDATE}(\nabla\mathcal{L}(\theta))$

---

ism of Pooladian et al. (2023) and Tong et al. (2023), the same samples $\mathbf{x}_0^1 \ldots, \mathbf{x}_0^n$ and $\mathbf{x}_1^1, \ldots, \mathbf{x}_1^n$ are first fed into a discrete optimal matching solver. This outputs a bistochastic coupling matrix $\mathbf{P} \in \mathbb{R}^{n \times n}$ which is then used to *re-shuffle* the $n$ pairs originally provided to be better coupled, and which should help the velocity field fit straighter trajectories, with less training steps. The procedure is summarized in Algorithm 2 and adapted to our setup and notations. The choice $\mathbf{I}_n/n$ corresponds to I-FM, as it would return the original untouched pairs $(\mathbf{x}_0^k, \mathbf{x}_1^k)$. Equivalently, I-FM would also be recovered if the coupling was the independent coupling $\mathbf{1}_{n \times n}/n^2$, up to the difference in carrying out stratified sampling (which would result in each noise/image observed once per mini-batch) or sampling with replacement. More recently, Davtyan et al. (2025) have proposed to keep a memory of that matching effort across mini-batches, by updating a large (of the size of the entire dataset) assignment permutation between noise and full-batch data that is locally refreshed with the output of the Hungarian method run on a small batch.

## 3 Prepping Sinkhorn for Large Batch Size and Dimension.

A crucial aspect of the batch-OT methodology is that in its current implementations, any effort done to pair data more carefully with noise is disconnected from the training of $\mathbf{v}_\theta$ itself. Indeed, as currently implemented, OT variants of FM can be interpreted as meta-dataloaders that do a selective pairing of noise and data, without considering $\theta$ at all in that pairing. In that sense, training and preparation of coupled noise/data pairs can be done independently. We exploit this decoupling throughout this section: we first motivate why large batch sizes and careful choices of $\varepsilon$ matter (§3.1), then describe the practical ingredients needed to scale up Sinkhorn (§3.2).

### 3.1 Motivation and Metrics

**On Using Large Batch Size and Selecting $\varepsilon > 0$.** The motivation to use larger batch sizes for Batch-OT lies in the fundamental bias introduced by using small batches in light of the OT curse of dimensionality (Chewi et al., 2024; Fatras et al., 2019), which cannot be traded off with more iterations on the flow matching loss. Specifically, we provide the following lower bound that characterizes the statistical hardness of optimal transport, and defer its proof to the Appendix A.1.

**Proposition 2.** *Suppose the support of $\mu_1$ has intrinsic dimension $r$, formalized in Assumption 5. Define the coupling $X_0, X_1 \sim \pi_n$ as follows: first draw $\mathbf{X}_0 \sim \mu_0^{\otimes n}$ and $\mathbf{X}_1 \sim \mu_1^{\otimes n}$, then sample $X_0, X_1 \sim \hat{\pi}_n(\mathbf{X}_0, \mathbf{X}_1)$ for any coupling rule $\hat{\pi}_n$ supported on $\mathbf{X}_0, \mathbf{X}_1$. Then, for any $\mathbf{x}_0 \in \mathbb{R}^d$,*

$$\text{Var}_{X_0,X_1 \sim \pi_n}(X_1 \mid X_0 = \mathbf{x}_0) \ge cn^{-2/r},$$

*where $c > 0$ is a constant depending only on $C$ and $r$ of Assumption 5.*

Note that the above proposition covers the case of using couplings that are supported on batches of noise and data, as in Algorithm 1. When $\mu_0$ admits a density, the conditional variance under exact OT would be zero. Thus, Proposition 2 shows the curse of dimensionality in learning optimal transport *with any high-dimensional data distribution $\mu_1$*, which is in contrast to minimax lower bounds (e.g. Chewi et al. (2024, Theorem 2.15)) that only show the hardness for *some* unknown pair of distributions. This generality is at the expense of limiting the (stochastic) coupling to be supported on $(\mathbf{X}_0, \mathbf{X}_1)$, which is the relevant setting for flow matching. This curse of dimensionality becomes milder under the *manifold hypothesis* where $r \ll d$, but still advocates for the use of large $n$.

The necessity of varying $\varepsilon$ is that this regularization can offset the bias between a regularized empirical OT matrix and its coupling measure counterpart, with favorable sample complexity (Genevay et al., 2018; Mena & Niles-Weed, 2019; Rigollet & Stromme, 2025).

**Automatic Rescaling of $\varepsilon$.** A practical problem arising when running the Sinkhorn algorithm lies in choosing the $\varepsilon$ parameter. As described earlier, while $\mathbf{P}^\varepsilon$ does follow a path from the optimal permutation (i.e., returned by the Hungarian algorithm) to the independent coupling, as $\varepsilon$ varies from 0 to $\infty$, what matters in practice is to pick relevant values in between these two extremes. To avoid using a fixed grid that risks becoming irrelevant as we vary $n$ and $d$, we revisit the strategy originally used in Cuturi (2013) to divide the cost matrix $\mathbf{C}$ by its mean, median or maximal value, as implemented for instance in Flamary et al. (2021). While needed to avoid underflow when instantiating a kernel matrix $\mathbf{K} = e^{-\mathbf{C}/\varepsilon}$, that strategy is not relevant when using the log-sum-exp operator in our implementation (as advocated in Peyré & Cuturi (2019, Remark 4.23)), since the $\min_\varepsilon$ in our implementation is *invariant* to a constant shift in $\mathbf{C}$, whereas mean, median and max statistics are not. We propose instead to use the *standard deviation* (STD) of the cost matrix. Indeed, the dispersion of costs around their mean has more relevance as a scale than the mean of these costs itself. The STD can be computed in $nd^2$ time / memory, without having to instantiate the cost matrix. When this memory cost increase from $d$ to $d^2$ is too high, we subsample $n = 2^{14} = 16{,}384$ points. In what follows, we always pass the $\varepsilon$ value to the Sinkhorn algorithm 1 as $\tilde{\varepsilon} := \text{std}(\mathbf{C}) \times \varepsilon$, where $\varepsilon$ is now a scale-free quantity selected in a logarithmic grid within $[0.001, 1.0]$.

**Scale-Free Renormalized Coupling Entropy.** While useful to keep computations stable across runs, the rescaling of $\varepsilon$ still does not provide a clear idea of whether a computed coupling $\mathbf{P}^\varepsilon$ from $n$ to $n$ points is sharp (close to an optimal permutation) or blurred (closer to what I-FM would use). While a distance to the independent coupling can be computed, that to the optimal Hungarian permutation cannot, of course, be derived without computing it beforehand which would incur a prohibitive cost. Instead, we resort to a fundamental information inequality used in Cuturi (2013): if $\mathbf{P}$ is a valid coupling between two marginal probability vectors $\mathbf{a}, \mathbf{b}$, then one has $\frac{1}{2}(H(\mathbf{a}) + H(\mathbf{b})) \leq H(\mathbf{P}) \leq H(\mathbf{a}) + H(\mathbf{b})$. As a result, for any $\varepsilon$, we define the *renormalized* entropy $\mathcal{E}$ of a coupling of $\mathbf{a}, \mathbf{b}$:

$$\mathcal{E}(\mathbf{P}) := \frac{2H(\mathbf{P})}{H(\mathbf{a}) + H(\mathbf{b})} - 1 \in [0, 1].$$

When $\mathbf{a} = \mathbf{b} = \mathbf{1}_n/n$, as considered in this work, this simplifies to $\mathcal{E}(\mathbf{P}) := H(\mathbf{P})/\log n - 1$. Independently of the size $n$ and $\varepsilon$, $\mathcal{E}(\mathbf{P}^\varepsilon)$ provides a simple measure of the proximity of $\mathbf{P}^\varepsilon$ to an optimal assignment matrix (as $\mathcal{E}$ gets closer to 0) or to the independent coupling (as $\mathcal{E}$ reaches 1). As a result we report $\mathcal{E}(\mathbf{P}^\varepsilon)$ rather than $\varepsilon$ in our plots (or to be more accurate, the *average* of $\mathcal{E}(\mathbf{P}^\varepsilon)$ computed over multiple mini-batches). Figures 9 and 10 in the appendix are indexed by $\varepsilon$ instead.

**From Squared Euclidean Costs to Dot-products.** Using the notation $T^\star(\mu, \nu)$ introduced in (2), we notice an equivariance property of Monge maps. For $\mathbf{s} \in \mathbb{R}^d$ and $r \in \mathbb{R}_+$ we write $L_{r,\mathbf{s}}$ for the dilation and translation map $L_{r,\mathbf{s}}(\mathbf{x}) = r\mathbf{x} + \mathbf{s}$. Naturally, $L_{r,\mathbf{s}}^{-1}(\mathbf{x}) = (\mathbf{x} - \mathbf{s})/r = L_{1/r, -\mathbf{s}/r}(\mathbf{x})$, but also $L_{r,s} = \nabla w_{r,s}$ where $w_{r,s}(\mathbf{x}) := \frac{r}{2}\|\mathbf{x}\|^2 - \mathbf{s}^T\mathbf{x}$ is convex.

**Lemma 3.** *The Monge map $T(\mu, \nu)$ is equivariant w.r.t to dilation and translation maps, as*

$$T^\star((L_{r,\mathbf{s}})_\# \mu, (L_{r',\mathbf{s}'})_\# \nu) = L_{r',\mathbf{s}'} \circ T^\star(\mu, \nu) \circ L_{r,\mathbf{s}}^{-1}.$$

*Proof.* Following Brenier's theorem, let $u$ be a convex potential such that $T^\star(\mu, \nu) = \nabla u$. Set $F := L_{r',\mathbf{s}'} \circ \nabla u \circ L_{r,\mathbf{s}}^{-1}$. Then $F$ is the composition of the gradients of 3 convex functions. Because the Jacobians of $L_{r,s}$ and $L_{r,\mathbf{s}}^{-1}$ are respectively $r\mathbf{I}_d$ and $\mathbf{I}_d/r$, they commute with the Hessian of $u$. Therefore the Jacobian of $F$ is symmetric, positive definite, and $F$ is the gradient of a convex potential that pushes $(L_{r,\mathbf{s}})_\# \mu$ to $(L_{r',\mathbf{s}'})_\# \nu$, and is therefore their Monge map by Brenier's theorem. $\square$

In practice this equivariance means that, when focusing on permutation matrices (which can be seen as the discrete counterparts of Monge maps), one is free to rescale and shift either point cloud. This remark has a practical implication when running Sinkhorn as well. When using the squared-Euclidean distance matrix, the cost matrix is a sum of a correlation term with two rank-1 norm terms, $\mathbf{C} = -\mathbf{X}\mathbf{Y}^T + \frac{1}{2}(\boldsymbol{\xi}\mathbf{1}_n^T + \mathbf{1}_n\boldsymbol{\gamma}^T)$

where $\boldsymbol{\xi}$ and $\boldsymbol{\gamma}$ are the vectors composed of the $n$ squared norms of vectors in $\mathbf{X}$ and $\mathbf{Y}$. Yet, due to the constraints $\mathbf{P1}_n = \mathbf{a}, \mathbf{P}^T \mathbf{1}_n = \mathbf{b}$, any modification to the cost matrix of the form $\tilde{\mathbf{C}} = \mathbf{C} - \mathbf{c1}_n^T - \mathbf{1}_n \mathbf{d}^T$, where $\mathbf{c}, \mathbf{d} \in \mathbb{R}^n$ only shifts the (3) objective by a constant: $\langle \mathbf{P}, \tilde{\mathbf{C}} \rangle = \langle \mathbf{P}, \mathbf{C} \rangle - \frac{1}{n} \mathbf{1}_n^T \mathbf{c} - \frac{1}{n} \mathbf{1}_n^T \mathbf{d}$. In practice, this means that norms only perturb Sinkhorn without altering the optimal coupling, and one should focus on the negative correlation matrix $\mathbf{C} := -\mathbf{XY}^T$, replacing Line 2 in Algorithm 1. We do observe substantial stability gains of these properly rescaled costs when comparing two point clouds (see Appendix A.2).

## 3.2 Scaling Sinkhorn in Practice

**Warm-starting Sinkhorn.** Solving the EOT problem (3) from scratch for each new batch of noise-data pairs $(\mathbf{X}_0, \mathbf{X}_1)$ is generally unnecessarily costly, since the solution is discarded each time a new batch is drawn. For large batch sizes, we propose to use the OT solution to $i$th batch $(\mathbf{X}_0^{(i)}, \mathbf{X}_1^{(i)})$ by warm-starting Sinkhorn for the $(i+1)$th batch $(\mathbf{X}_0^{(i+1)}, \mathbf{X}_1^{(i+1)})$. Let $(\mathbf{f}^\star, \mathbf{g}^\star)$ be the optimal dual potentials for a given batch $(\mathbf{X}, \mathbf{Y})$. Then, these potentials can be extended to the continuous domain:

$$\mathbf{f}(\mathbf{x}) = \varepsilon \log \tfrac{1}{n} + \min_\varepsilon (\mathbf{C}(\mathbf{x}, \mathbf{y}_j) - \mathbf{g}_j),$$
$$\mathbf{g}(\mathbf{y}) = \varepsilon \log \tfrac{1}{n} + \min_\varepsilon (\mathbf{C}(\mathbf{x}_i, \mathbf{y}) - \mathbf{f}_i).$$

For a new batch $(\mathbf{X}', \mathbf{Y}')$, we use the above formula to initialize the potentials $(\mathbf{f}', \mathbf{g}')$, i.e. $(\mathbf{f}', \mathbf{g}') \leftarrow (\mathbf{f}(\mathbf{x}'_i)_i, \mathbf{g}(\mathbf{y}'_j)_j)$. Since (3) is strictly convex, the choice of initialization has no influence on the solution. In practice, we find that warm-starting Sinkhorn substantially reduces the number of iterations required and the overall runtime of OTFM. We ablate the role of warmstart in Appendix A.3.

**Computing Matchings in PCA Space.** With the dot-product cost we can further use Principal Component Analysis (PCA) to optimally reduce the dimensionality of the cost matrix and significantly speed up Sinkhorn computation. Let $\mathbf{x}$ and $\mathbf{y}$ represent noise and data samples respectively, and let $\mathbf{A} \in \mathbb{R}^{k \times d}$ denote the projection matrix whose rows contain top-$k$ PCA directions. The PCA reconstruction of $\mathbf{y}$ is $\mathbf{A}^T \mathbf{Ay}$, and

$$\mathbf{x}^T \mathbf{y} \approx \mathbf{x}^T \mathbf{A}^T \mathbf{Ay} = \bar{\mathbf{x}}^T \bar{\mathbf{y}},$$

where $\bar{\mathbf{x}}$ and $\bar{\mathbf{y}}$ are the projection of $\mathbf{x}$ and $\mathbf{y}$ onto the PCA subspace. Note that we can achieve this dimensionality reduction regardless of the structure of $\mathbf{x}$, and this trick can be applied in the generative setting where $\mathbf{x}$ is an isotropic Gaussian vector. This reduces the naive runtime of computing the cost matrix from $n^2 d$ to $n^2 k$. For large $n$, we compute the cost matrix on the fly per Sinkhorn iteration and avoid materializing the entire matrix at once, hence this reduction also occurs per iteration. In our experiments, we can achieve an almost 10x speedup in Sinkhorn computation from PCA by using $k$ as small as 500 for the ImageNet-64 dataset with $d = 12,228$, without sacrificing generation quality; see Appendix A.4 for details.

**Precomputing Noise/Data Pairs.** We can completely separate the computational cost of preparing coupled noise/data pairs from the cost of training the model. To do so, as $n$ datapoints are retrieved from a dataloader and $n$ Gaussian samples are drawn, we can accumulate and buffer the outputs of Steps 1–4 of Algorithm 2 in a new augmented dataloader. To avoid storing noise vectors, we generate each noise vector using a single Pseudo-Random Number Generator (PRNG) key, and only store pairs of data identifier and the corresponding PRNG key for the coupled noise vector. When training an FM model (Steps 5–8 of Algorithm 2), we load pairs of data identifier and PRNG key from this new dataloader, retrieve the corresponding data, and generate the noise using the key. We use this approach while ablating any hyperparameters of FM training, to avoid Sinkhorn recomputations.

**Scaling Up Sinkhorn to Millions of High-Dimensional Points.** When guiding flow matching with batch-OT as presented in Algorithm 2, our ambition is to vary $n$ and $\varepsilon$ so that the coupling $\mathbf{P}^\varepsilon$ used to sample indices can be both large ($n \approx 10^6$) and sharp if needed, i.e. with an $\varepsilon$ that can be brought to arbitrarily low levels so that $\mathcal{E}(\mathbf{P}^\varepsilon) \approx 0$. To that end, we leverage the OTT-JAX implementation of the Sinkhorn algorithm (Cuturi et al., 2022), which can be natively sharded across multi-GPUs, or more generally multiple nodes of GPU machines equipped with efficient interconnect. In that approach, inspired by the earlier mono-GPU implementation of Feydy (2020), all $n$ points from source and target are sharded across GPUs and nodes (we have used either 1 or 2 nodes of 8 GPUs each, either NVIDIA H100 or A100). A crucial point in our implementation is that the cost matrix $\mathbf{C} = -\mathbf{XY}^T$ (following remark above) is never instantiated globally.

Instead, it is recomputed at each $\min_\varepsilon$ operation in Lines 4 and 5 of Algorithm 1 locally, for these shards. All sharded results are then gathered to recover $\mathbf{f}, \mathbf{g}$ newly assigned after that iteration. When outputted, we use $\mathbf{f}^\varepsilon$ and $\mathbf{g}^\varepsilon$ and, analogously, never instantiate the full $\mathbf{P}^\varepsilon$ matrix (this would be impossible at sizes $n \approx 10^6$ we consider) but instead, materialize it blockwise to do stratified index sampling corresponding to Line 4 in Algorithm 2. We use the Gumbel-softmax trick to vectorize the categorical sampling of each of these lines to select, for each line index $i$, the corresponding column $j_i$.

## 4 Experiments

We revisit the application of Algorithm 2 using the modifications to the Sinkhorn algorithm outlined in Section 3 to consider various benchmark tasks for which I-FM has been used. We consider synthetic tasks in which the ground-truth Monge map is known, and benchmark unconditioned image generation using CIFAR-10 (Krizhevsky et al., 2009), and the 32×32 and 64×64 downsampled variants (Chrabaszcz et al., 2017) of the ImageNet dataset (Deng et al., 2009).

**Sinkhorn Hyperparameters.** To track accurately whether the Sinkhorn algorithm converges for low $\varepsilon$ values, we set the maximal number of iterations to 50,000. We use the adaptive momentum rule introduced in Lehmann et al. (2022) beyond 2000 iterations to speed-up harder runs. Overall, all of the runs below converge: even for low $\varepsilon$, we achieve convergence except in a very few rare cases. The threshold $\tau$ is set to 0.001 and we observe that it remains relevant for all dimensions, as we use the 1-norm to quantify convergence.

### 4.1 Evaluation Metrics To Assess the Quality of a Flow Model $\mathbf{v}_\theta$

All metrics used in our experiments can be interpreted as *lower is better*.

**Negative log-likelihood.** Given a trained flow model $\mathbf{v}_\theta(t, \mathbf{x})$, the density $p_t(\mathbf{x})$ obtained by pushing forward $p_0(\mathbf{x})$ along the flow map of $\mathbf{v}_\theta$ can be computed by solving

$$\log p_t(\mathbf{x}_t) = \log p_0(\mathbf{x}_0) - \int_0^1 (\nabla_x \cdot \mathbf{v}_\theta)(t, \mathbf{x}_t)\, \mathrm{d}t, \qquad \dot{\mathbf{x}}_t = \mathbf{v}_\theta(t, \mathbf{x}_t), \tag{4}$$

Similarly, given a pair $(t, \mathbf{x})$, the density $p_t(\mathbf{x})$ can be evaluated by backward integration (Grathwohl et al., 2018, Section 2.2). The divergence $(\nabla_x \cdot \mathbf{v}_\theta)(t, \mathbf{x}_t)$ requires computing the trace of the Jacobian of $\mathbf{v}_\theta(t, \cdot)$. As commonly done in the literature, we use the Hutchinson trace estimator with a varying number of samples to speed up that computation without materializing the entire Jacobian and use either an `Euler` solver with 50 steps for synthetic tasks or a `Dopri5` adaptive solver for image generation tasks, both implemented in the Diffrax toolbox (Kidger, 2021). Given $n$ points $\mathbf{x}_1^1, \ldots, \mathbf{x}_1^n \sim \nu$, the negative log-likelihood (NLL) of that set is

$$\mathcal{L}(\theta) := -\frac{1}{n} \sum_{i=1}^n \log p_1(\mathbf{x}_1^i).$$

subject to (4). We alternatively report the bits per dimension (BPD) statistic, given by $\mathrm{BPD} = \mathcal{L}/(d \log 2)$.

**Curvature.** We use the *curvature* of the field $\mathbf{v}_\theta$ as defined by Lee et al. (2023): for $n$ integrated trajectories $(\mathbf{x}_t^1, \ldots, \mathbf{x}_t^n)$ starting from samples at $t = 0$ from $\mu$, the curvature is defined as

$$\kappa(\theta) := \frac{1}{n} \sum_{i=1}^n \int_0^1 \|\mathbf{v}_\theta(t, \mathbf{x}_t^i) - (\mathbf{x}_1^i - \mathbf{x}_0^i)\|_2^2 \mathrm{d}t,$$

where the integration is done with an `Euler` solver with 50 steps for synthetic tasks and the `Dopri5` solver evaluated on a grid of 8 steps for image generation tasks. The smaller the curvature, the more the ODE path looks like a straight line, and should be easy to integrate.

**Reconstruction loss.** For synthetic tasks in Sections 4.2, we have access to the ground-truth transport map $T_0$ that generated the target measure $\mu_1$ from $\mu_0$. In both cases, that map is parameterized as the gradient of a convex Brenier potential, respectively a piecewise quadratic function and an input convex neural network, ICNN (Amos et al., 2017). For a starting point $\mathbf{x}_0$, we can therefore compute a *reconstruction loss* (a variant of the $\mathcal{L}^2$-UVP in Korotin et al. (2021)) as the squared norm of the difference between the true map $T^\star(\mathbf{x}_0)$

and the flow map $T_\theta$ obtained by integrating $\mathbf{v}_\theta(t, \cdot)$ (using a varying number of steps with a `Euler` solver or with the `Dopri5` solver), defined using $n$ points sampled from $\mu$ as

$$\mathcal{R}(\theta) := \frac{1}{n} \sum_{i=1}^{n} \|T_\theta(\mathbf{x}_0^i) - T_0(\mathbf{x}_0^i)\|_2^2.$$

**FID.** We report the FID metric (Heusel et al., 2017) in image generation tasks. For CIFAR-10 we use the train dataset of 50,000 images, for ImageNet-32 and ImageNet-64 we subset a random set of 50,000 images from the train set. For generation we consider four integration solvers, `Euler` with 4, 8 and 16 steps (a.k.a. Number of Function Evaluations (NFE)) and a `Dopri5` solver from the Diffrax library (Kidger, 2021).

### 4.2 Synthetic Benchmark Tasks, $d = 32 \sim 256$

We consider in this section synthetic benchmarks of medium dimensionality ($d = 32 \sim 256$). We favor this synthetic setting over other data sources with similar dimensions (e.g. PCA reduced single-cell data (Bunne et al., 2024)) in order to have access to the ground-truth reconstruction loss, which helps elucidate the impact of OT batch size $n$ and $\varepsilon$.

**Piecewise Affine Brenier Map.** The source is a standard Gaussian and the target is obtained by mapping it through the gradient of a potential, itself a (convex) piecewise quadratic function obtained using the pointwise maximum of $k$ rank-deficient parabolas:

$$u(\mathbf{x}) := \max_{i \leq k} u_i(\mathbf{x}) := \tfrac{1}{2}\|\mathbf{x}\|^2 + \tfrac{1}{2}\|\mathbf{A}_i(\mathbf{x} - \mathbf{m}_i)\|^2 - \|\mathbf{A}_i\mathbf{m}_i\|^2, \tag{5}$$

where $\mathbf{A}_i \sim \text{Wishart}(\frac{d}{2}, I_d), \mathbf{m}_i \sim \mathcal{N}(0, 3I_d), c_i \sim \mathcal{N}(0, 1)$ and all means are centered around zero after sampling. In practice, this yields a transport map of the form $\nabla u(\mathbf{x}) = \mathbf{x} + \mathbf{A}_{i^\star}(\mathbf{x} - \mathbf{m}_{i^\star})$ where $i^\star$ is the potential selected for that particular $\mathbf{x}$ (i.e. the argmax in (5)). The correction $-\|\mathbf{A}_i\mathbf{m}_i\|^2$ is designed to ensure that these potentials are sampled equally even when $\mathbf{m}_i$ is sampled far from 0. The number of potentials $k$ is set to $d/16$. Examples of this map are shown in Appendix A.5 for dimension 128. We consider this setting in dimensions $d = 32, 64, 128, 256$.

**Korotin et al. Benchmark.** The source is a predefined Gaussian mixture and the ground-truth OT map is a pre-trained ICNN. We consider this benchmark in various dimensions $d = 32 \sim 256$, using their ICNN checkpoints (see Appendix A.6). This problem is more challenging than the previous one, because both the source *and* target distributions have multiple modes, and the OT map itself is a fairly complex ICNN.

**Velocity Field Parameterization and Training.** The velocity fields are parameterized as MLPs with 5 hidden layers, each of size 512 when $d = 32, 64$ and 1024 when $d = 128, 256$. Time in $[0, 1]$ is encoded using $d/8$ Fourier encodings. All models are trained with unpaired batches: the sampling in Line 1 of Algorithm 2 is done as $\mathbf{X}_0 \sim \mu$ while for Line 2, $\mathbf{X}_1 := T_0(\mathbf{X}_0')$ where $\mathbf{X}_0'$ is a new sample from $\mu$ and $T$ is applied to each of the $n$ points described in $\mathbf{X}_0'$. All models are trained for 8192 steps, with effective batch sizes of 2048 samples (256 per GPU) to average a gradient, a learning rate of $10^{-3}$ (we tested with $10^{-2}$ or $10^{-4}$, the former was unstable while the latter was less efficient on a subset of runs). The model marked as ▲ in the plots is a flow model trained with *perfect* supervision, i.e. given *ground-truth paired samples* $\mathbf{X}_0 \sim \mu$ and $\mathbf{X}_1 := T_0(\mathbf{X}_0)$, provided in the correct order. I-FM is marked as

| NFE | 4 | 8 | 16 | Adaptive |
|---|---|---|---|---|
| I-FM | | | | |
| $n = 2^{12}$ | | | | |
| $n = 2^{17}$ | | | | |

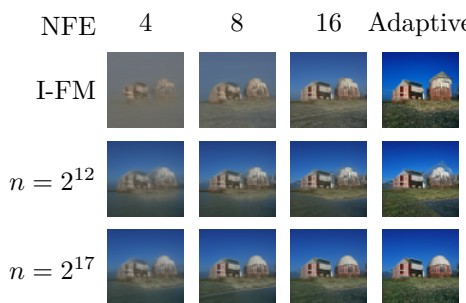

Figure 1: Samples generated from models trained on **ImageNet-64**. $n$ denotes the total OT batch size. We use $\varepsilon = 0.1$ and the `Euler` solver (`Dopri5` for adaptive with NFE $\approx 270$). More samples provided in Figure 14.

▼. For all other runs, we vary $\varepsilon$ (reporting renormalized entropy $\mathcal{E}(\mathbf{P}^\varepsilon)$) and the total batch size $n$ used to compute couplings, somewhere between 2048 and 2,097,152. These runs are carried out on a single node with 8 GPUs, and therefore the data is sharded in blocks of size $n/8$ when running the Sinkhorn algorithm.

**Results.** The results displayed in Figures 2 and 3 paint a homogeneous picture: as can be expected, increasing $n$ is generally impactful and beneficial for all metrics. The interest of decreasing $\varepsilon$, while beneficial

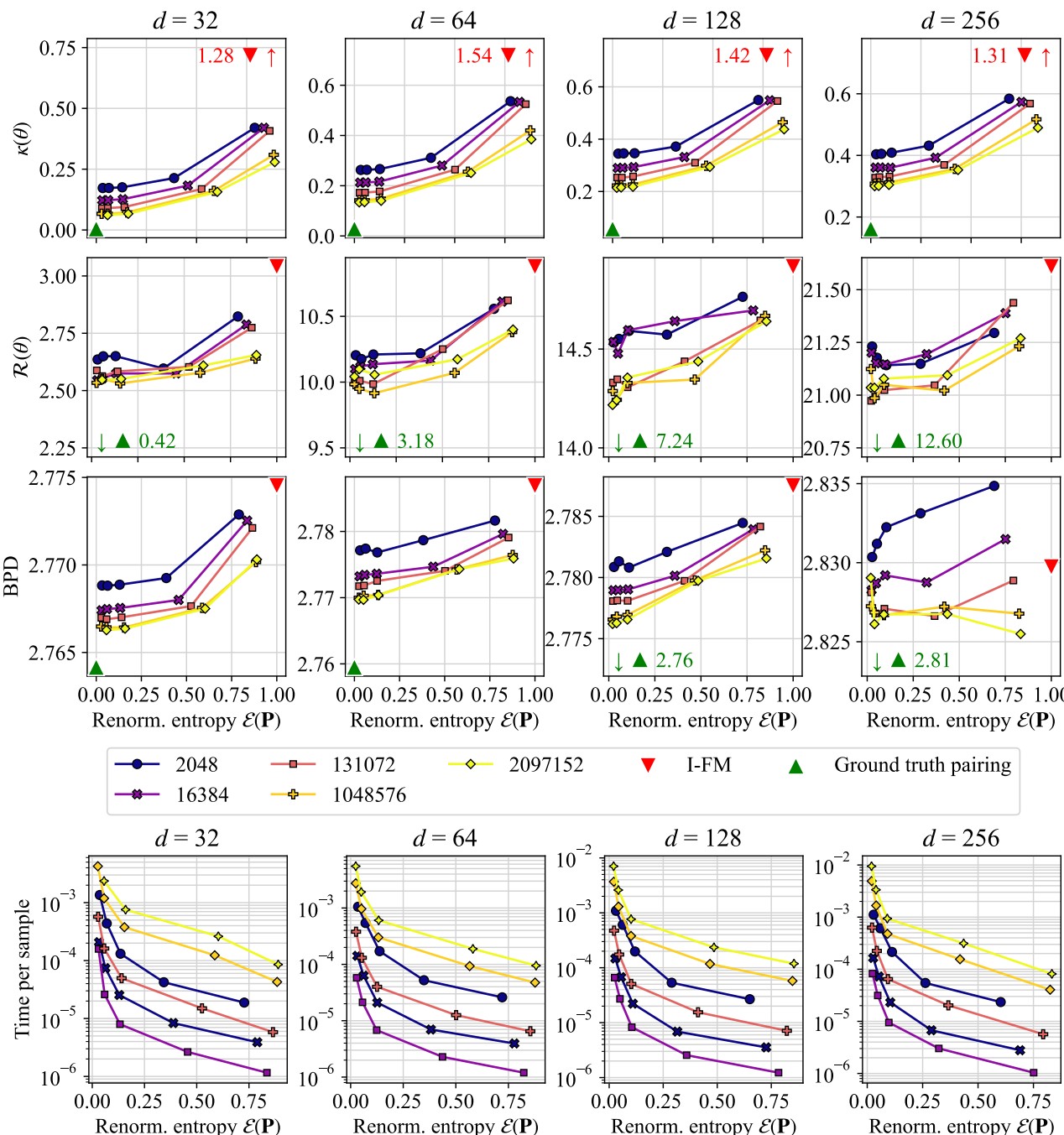

Figure 2: Results on the **piecewise affine OT Map benchmark**. The three top rows present (in that order) curvature, reconstruction and BPD metrics. Below, we provide compute times associated with running the Sinkhorn algorithm as a per-example cost. This per-example cost is the total time needed to run Sinkhorn to get $n \times n$ coupling divided by $n$. That cost would be 0 when using I-FM. We observe across all dimensions improvements of all metrics.

in smaller dimensions, can be less pronounced in higher dimensions. Indeed, we find that renormalized entropies around $\approx 0.1$ should be advocated, if one has in mind the computational effort needed to get these samples, pictured at the bottom of each figure.

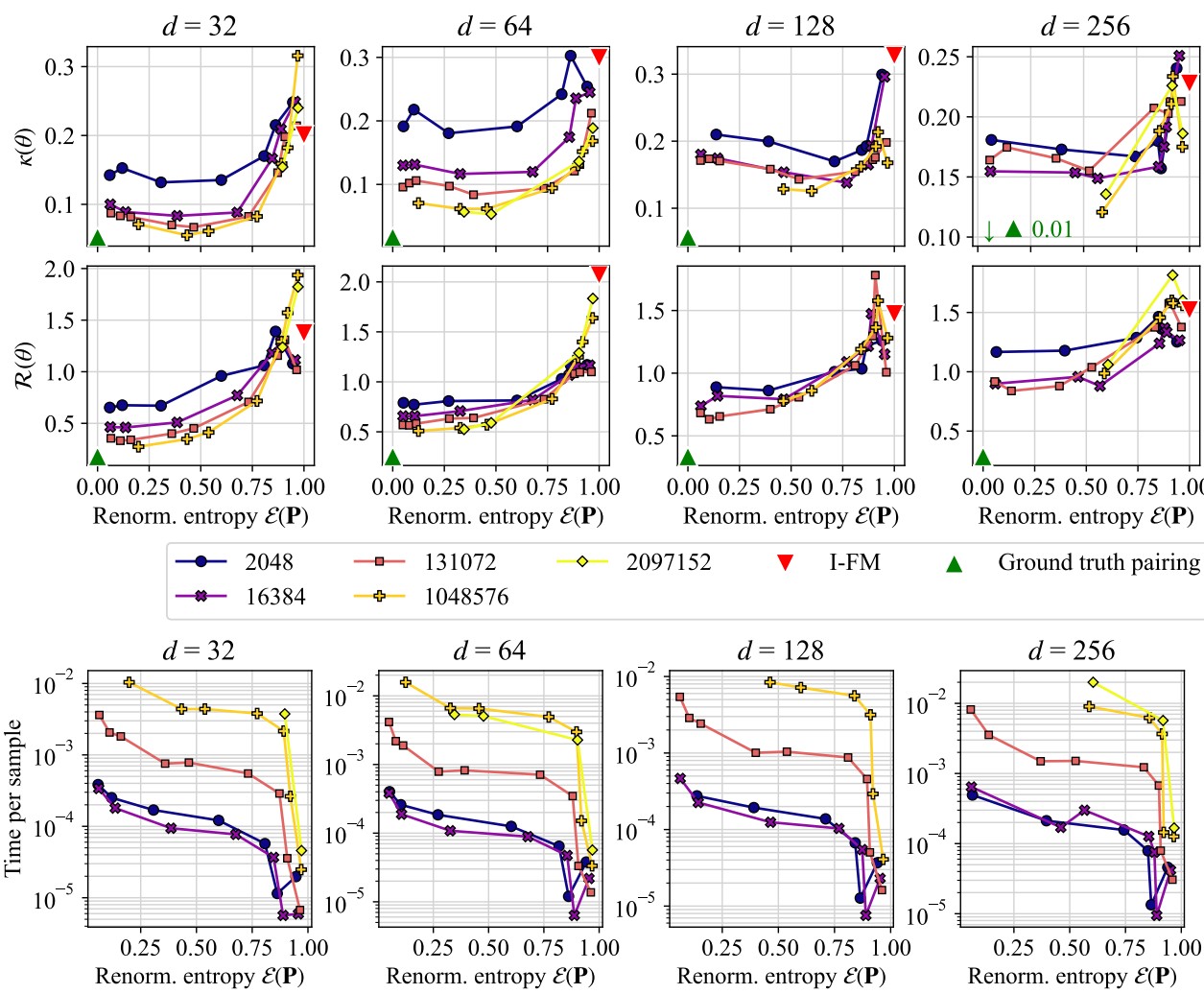

Figure 3: Results on the **Korotin benchmark**. As with Figure 2, we compute curvature and reconstruction metrics, and compute times below. Some of the runs for largest OT batch sizes $n$ are provided in the supplementary. These runs suggest that to train OT models in these dimensions increasing $n$ is overall beneficial across the board.

| | ImageNet-32 | | | | ImageNet-64 | | | |
|---|---|---|---|---|---|---|---|---|
| NFE → 
 $n$ ↓ | 4 | 8 | 16 | Adaptive 
 $115 \pm 1$ | NFE → 
 $n$ ↓ | 4 | 8 | 16 | Adaptive 
 $269 \pm 1$ |
| I-FM | 66.4 | 24.3 | 12.1 | 5.55 | I-FM | 80.1 | 37.0 | 19.5 | 9.32 |
| 2048 | 38.2 | 16.8 | 10.0 | 5.89 | 4096 | 50.3 | 25.0 | 15.8 | 9.39 |
| 65536 | 33.1 | 15.1 | 9.28 | 4.88 | 32768 | 48.8 | 24.6 | 15.7 | 9.08 |
| 524288 | **31.5** | **14.8** | **9.19** | **4.85** | 131072 | **46.9** | **23.9** | **15.4** | **8.99** |

Table 1: FID for models trained across different OT batch sizes. We use the best checkpoint (w.r.t FID at `Dopri5`) for each model, restricting results to the setting where the relative `epsilon` value $\varepsilon = 0.1$ for ease of presentation (more detailed results can be seen in the plots of Figure 5).

### 4.3 Unconditioned Image Generation, $d = 3072 \sim 12288$.

As done originally in Lipman et al. (2023), we consider unconditional generation of the CIFAR-10, ImageNet-32 and ImageNet-64 datasets.

**Velocity Field Parameterization and Training.** We use the network parameterization given in Tong et al. (2024, see Section E.8) for CIFAR-10 and those given in Pooladian et al. (2023, see Table 10) for ImageNet-32 and ImageNet-64. We follow their recommendations on setting learning rates, batch sizes (to average gradients) as well as total number of iterations: we train respectively for 400k, 438k and 957k using effective batch sizes advocated in their paper, respectively $16 \times 8$, $128 \times 8$ and $50 \times 16$. We summarize these choices in Table 4.

**CIFAR-10.** Results are presented in Figure 4, and further details in Appendix A.7. Compared to results reported in Tong et al. (2023) we observe slightly better FID scores (about 0.1) for both I-FM and Batch OT-FM. Note that the size of the dataset itself (50k, 100k when including random flipping as we do) is comparable (if not slightly lower) to our largest batch size $n = 131{,}072$, meaning some images are duplicated. Overall, the results show the benefit of relatively larger batch sizes and suitably small $\varepsilon$, that is more pronounced at lower NFE.

**ImageNet-32 and ImageNet-64.** Results are shown in Figures 5 and 6, and further details in Appendices A.8 and A.9. Compared to results reported in Tong et al. (2023) we observe slightly better FID scores (about 0.1 when using the `Dopri5` solver for instance) for I-FM. Compared to CIFAR-10, these datasets are more suitable for our large OT batch sizes as they contain significantly more samples, and we continue to observe the benefits of larger batch size and proper choice of renormalized entropy.

### Conclusion

Our experiments show that guiding flow models with large-scale Sinkhorn couplings is beneficial for downstream performance. We have tested this hypothesis by computing and sampling from both crisp and blurry $n \times n$ Sinkhorn coupling matrices for sizes $n$ in the millions of points, placing them on an intuitive scale from 0 (close to using an optimal permutation as returned e.g. by the Hungarian algorithm) to 1 (equivalent to the independent sampling approach popularized by Lipman et al. (2023)). This involved efficient multi-GPU parallelization, realizing scales which, to our knowledge, were never achieved previously in the literature. Although the scale of these computations may seem large, they are still relatively cheap compared to the cost of optimizing the FM loss, and are completely independent from model training. As a result, they should be carried out prior to any training.

For mid-sized problems, paying this relatively small price to compute and sample paired indices from large-scale couplings yields substantial returns: faster training and faster inference, thanks to the straightness of the flows learned with the batch-OT procedure. For larger-sized problems, the picture is more nuanced, although we observe clear benefits when using batch sizes in the thousands, with further improvements when going beyond hundreds of thousands, and renormalized entropies of around 0.1. This forms our main practical recommendation for end users.

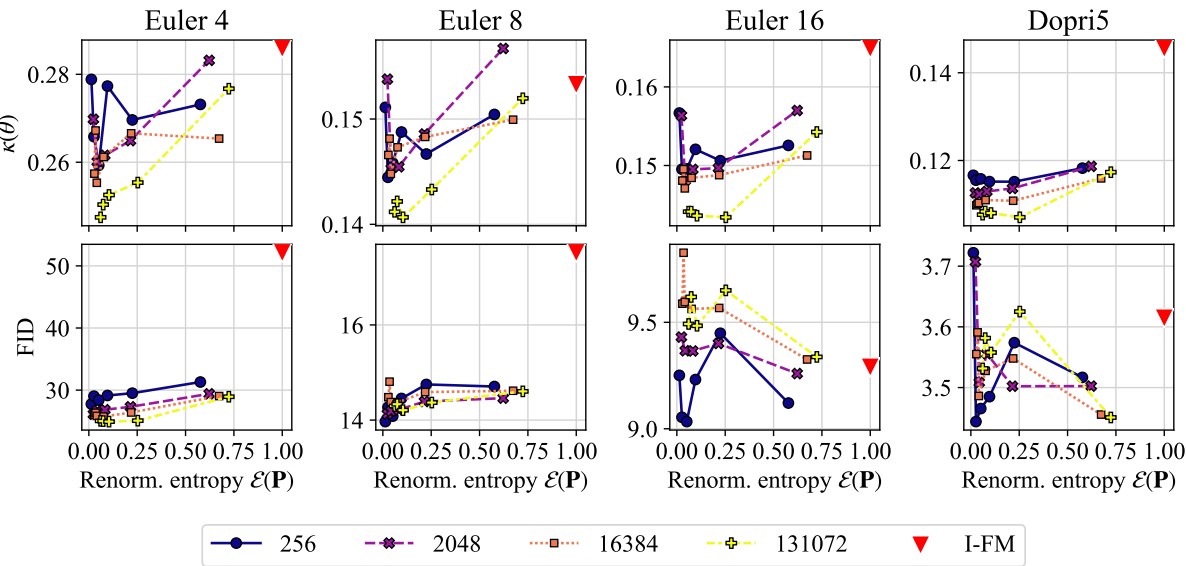

Figure 4: Experiment metrics for **CIFAR-10** image generation. We evaluate the trained models using the `Euler` solver with three different number of steps, and with the `Dopri5` solver and adaptive steps. The plots demonstrate the benefits of a larger OT batch size to achieve significantly smaller curvature, and moderately smaller FID at low number of integration steps. CIFAR-10 is not necessarily the best setup to evaluate the performance of OT based FM, since the number of points is relatively low (the batch sizes we consider involve in fact resampling *data*). Our experiments also suggest that in this setting, lower renormalized entropy generally benefits the performance.

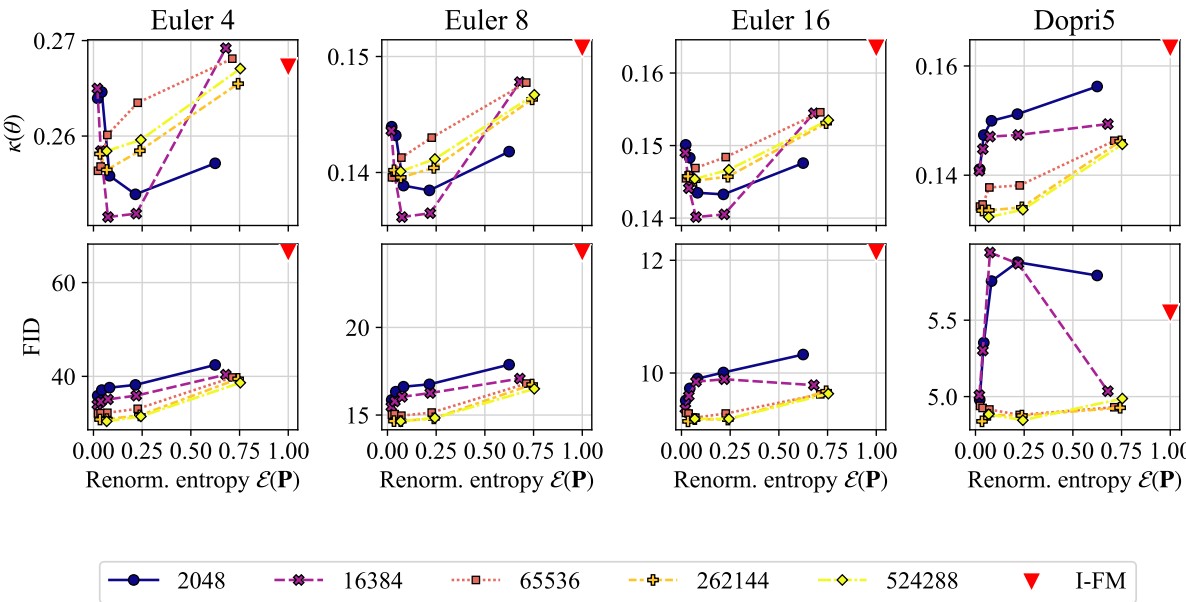

Figure 5: **ImageNet-32** experiment metrics. We observe that both FID and curvature are smaller when using larger OT batch size, and smaller renormalized entropy tends to result in better metrics.

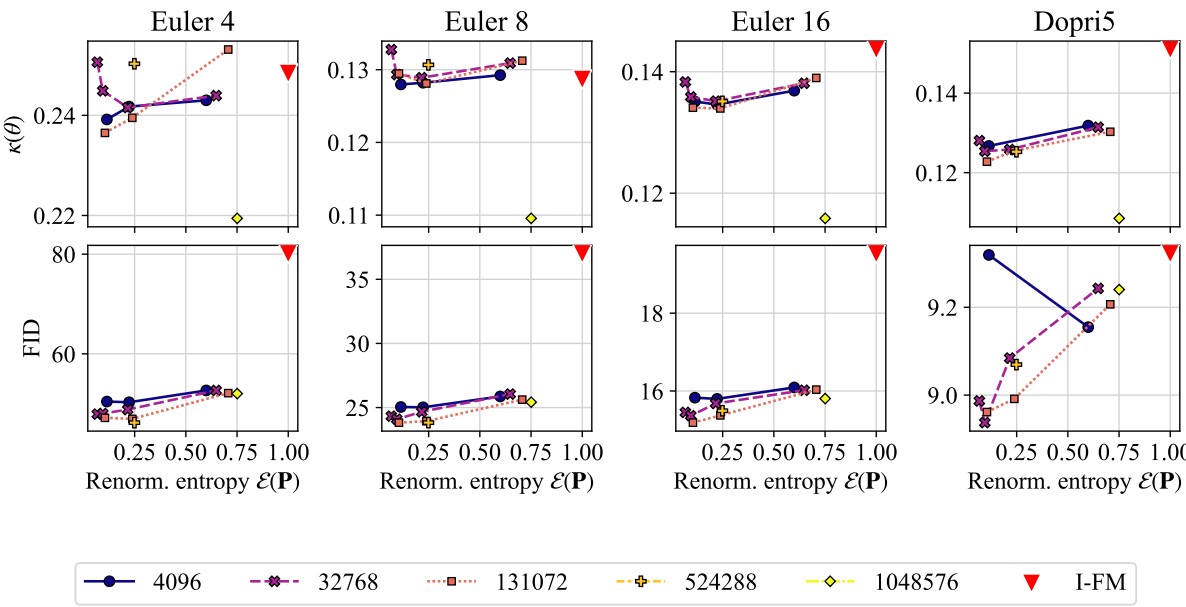

Figure 6: **ImageNet-64** results: Curvature and FID obtained with `Euler` integration with varying number of steps, as well as `Dopri5` integration.

Finally, we note that in a recent follow-up work (Mousavi-Hosseini et al., 2026), we explore flow matching using the *semidiscrete* formulation of OT (Genevay et al., 2018) as an alternative to using minibatches with Sinkhorn as we suggest in the present work. Compared to minibatch OT, the semidiscrete formulation of OT is tailored to the setting where the source distribution is continuous and the target distribution is finite, and this is the typical setting for generative modeling in practice. Semidiscrete OT treats the source distribution as being fully continuous and avoids minibatches, instead solving for a single global transport potential. Assuming that one can solve for this transport potential to sufficient accuracy, arbitrary noise samples can be coupled to target data points in linear time, while potentially avoiding the bias associated with discretizing the continuous source distribution.

**Limitations.** Our results rely on training of neural networks. In the interest of comparison, we have used the same model across all changes advocated in the paper (on $n$ and $\varepsilon$). However, and due to the scale of our experiments, we have not been able to ablate important parameters such as learning rates when varying $n$ and $\varepsilon$, and instead relied on those previously proposed for I-FM.

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

# A   Appendix

## A.1   The Necessity of Large OT Batch Size

Here, we formalize the assumptions in and provide the proof of Proposition 2.

Assuming that $\mu_0$ admits a density, if we were to couple $X_0$ and $X_1$ through optimal transport, by Theorem 1 we would have $\mathrm{Var}(X_1 \,|\, X_0) = 0$ a.s. over $X_0$, where variance is the sum of coordinate variances. In general, any coupling that provides $\mathrm{Var}(X_1 \,|\, X_0) = 0$ allows for one-step generation, simply by performing least-squares regression to learn $\mathbb{E}[X_1 \,|\, X_0]$. Therefore, we adopt $\mathrm{Var}(X_1 \,|\, X_0)$ as a measure of success of a coupling.

Recall from 2 that to obtain a pair of samples $X_0, X_1$ for training, we first draw n i.i.d. samples $\mathbf{X}_0 \sim \mu_0^{\otimes n}$ and $\mathbf{X}_1 \sim \mu_1^{\otimes n}$. Then, we sample $X_0, X_1 \sim \hat{\pi}_n(\mathbf{X}_0, \mathbf{X}_1)$, where $\hat{\pi}_n$ denotes the discrete optimal (entropic) transport solution between the uniform distribution on $\mathbf{X}_0$ and $\mathbf{X}_1$. We only require $\hat{\pi}_n(\mathbf{X}_0, \mathbf{X}_1)$ to be supported on $\mathbf{X}_0 \times \mathbf{X}_1$, as formalized by the following assumption.

**Assumption 4.** $\hat{\pi}_n(\mathbf{X}_0, \mathbf{X}_1)$ is supported on $\mathbf{X}_0$ and $\mathbf{X}_1$, more precisely,

$$\hat{\pi}_n(\mathbf{X}_0, \mathbf{X}_1) = \sum_{i,j}^{n} P_{ij}(\mathbf{X}_0, \mathbf{X}_1)\delta_{(\mathbf{x}_0^{(i)}, \mathbf{x}_1^{(j)})},$$

where $(P_{ij}(\mathbf{X}_0, \mathbf{X}_1))_{ij}$ is some bistochastic matrix, equivariant under permutations of $\mathbf{X}_0$ and $\mathbf{X}_1$, and $\delta$ denotes the Dirac measure.

To capture the intrinsic dimension of data, we can impose the following assumption on $\mu_1$.

**Assumption 5.** For $X$ and $X'$ drawn independently from the data distribution $\mu_1$, we have

$$\mathbb{P}[\|X - X'\| \leq t] \leq Ct^r,$$

for all $t > 0$ and some $C, r > 0$.

Note that the volume of an $r$-dimensional ball of radius $t$ is proportional to $t^r$. Therefore, $r$ in the above assumption roughly captures the intrinsic dimension of data, typically assumed to be much less than the ambient dimension, i.e. $r \ll d$.

We are now ready to present the proof of Proposition 2, which we repeat here for ease of reference.

**Proposition 6.** *Suppose $\hat{\pi}_n$ is any coupling rule that satisfies Assumption 4, and that $\mu_1$ satisfies Assumption 5. Define the coupling $X_0, X_1 \sim \pi_n$ as follows: first draw $\mathbf{X}_0 \sim \mu_0^{\otimes n}$ and $\mathbf{X}_1 \sim \mu_1^{\otimes n}$, then sample $X_0, X_1 \sim \hat{\pi}_n(\mathbf{X}_0, \mathbf{X}_1)$. Then, for any $\mathbf{x}_0 \in \mathbb{R}^d$, we have*

$$\mathrm{Var}_{X_0, X_1 \sim \pi_n}(X_1 \,|\, X_0 = \mathbf{x}_0) \geq cn^{-2/r},$$

*where $c > 0$ is a constant depending only on $C$ and $r$.*

To prove Proposition 6, we use the fact that

$$\mathrm{Var}(X_1 \,|\, X_0 = \mathbf{x}_0) = \frac{1}{2}\mathbb{E}[\|X_1 - X_1'\|^2 \,|\, X_0 = \mathbf{x}_0],$$

where $X_1$ and $X_1'$ are drawn independently from $\pi_n(\cdot \,|\, X_0 = \mathbf{x}_0)$. However, $X_1$ and $X_1'$ essentially come from different batches $\mathbf{X}_1$ and $\mathbf{X}_1'$, and their only dependence is through being coupled with $X_0 = \mathbf{x}_0$. We can remove this dependence by lower bounding the variance by the minimum distance between two batches of i.i.d. samples $\mathbf{X}_1$ and $\mathbf{X}_1'$. This is performed by the following lemma.

**Lemma 7.** *Let $\pi_n$ be as defined in Proposition 6. Then, for any $\mathbf{x}_0 \in \mathbb{R}^d$, we have*

$$\mathrm{Var}_{X_0, X_1 \sim \pi_n}(X_1 \,|\, X_0 = \mathbf{x}_0) \geq \frac{1}{2}\mathbb{E}_{\mathbf{X}_1, \mathbf{X}_1' \sim \mu_1^{\otimes n} \otimes \mu_1^{\otimes n}}[D(\mathbf{X}_1, \mathbf{X}_1')],$$

*where $D(\mathbf{X}_1, \mathbf{X}_1') := \min_{\mathbf{x}_1 \in \mathbf{X}_1, \mathbf{x}_1' \in \mathbf{X}_1'} \|\mathbf{x}_1 - \mathbf{x}_1'\|^2$.*

*Proof.* To draw $X_1, X_1' \sim \pi_n(\cdot \mid X_0 = \mathbf{x}_0) \otimes \pi_n(\cdot \mid X_0 = \mathbf{x}_0)$ we can draw $\mathbf{X}_0, \mathbf{X}_0' \sim \mu_0^{\otimes n} \otimes \mu_0^{\otimes n}$ and $\mathbf{X}_1, \mathbf{X}_1' \sim \mu_1^{\otimes n} \otimes \mu_1^{\otimes n}$. We then replace the first sample in $\mathbf{X}_0$ and $\mathbf{X}_0'$ with $\mathbf{x}_0$ to condition on $X_0 = \mathbf{x}_0$ (in particular, we rely on the equivariance of $\hat{\pi}_n$), and denote them by $\mathbf{X}_0(\mathbf{x}_0)$ and $\mathbf{X}_0'(\mathbf{x}_0)$. We can then write

$$
\begin{aligned}
\mathrm{Var}(X_1 \mid X_0 = \mathbf{x}_0) &= \frac{1}{2}\mathbb{E}[\|X_1 - X_1'\|^2 \mid X_0 = \mathbf{x}_0] \\
&= \frac{1}{2}\mathbb{E}\big[\mathbb{E}[\|X_1 - X_1'\|^2 \mid \mathbf{X}_0(X_0), \mathbf{X}_0'(X_0), \mathbf{X}_1, \mathbf{X}_1'] \mid X_0 = \mathbf{x}_0\big] \\
&= \frac{1}{2}\mathbb{E}\left[\sum_{\mathbf{x}_1 \in \mathbf{X}_1, \mathbf{x}_1' \in \mathbf{X}_1'} \hat{\pi}_n(\mathbf{X}_0(\mathbf{x}_0), \mathbf{X}_1)(\mathbf{x}_1 \mid \mathbf{x}_0)\hat{\pi}_n(\mathbf{X}_0'(\mathbf{x}_0), \mathbf{X}_1')(\mathbf{x}_1' \mid \mathbf{x}_0)\|\mathbf{x}_1 - \mathbf{x}_1'\|^2\right] \\
&\geq \frac{1}{2}\mathbb{E}\left[D(\mathbf{X}_1, \mathbf{X}_1')\sum_{\mathbf{x}_1 \in \mathbf{X}_1, \mathbf{x}_1' \in \mathbf{X}_1'} \hat{\pi}_n(\mathbf{X}_0(\mathbf{x}_0), \mathbf{X}_1)(\mathbf{x}_1 \mid \mathbf{x}_0)\hat{\pi}_n(\mathbf{X}_0'(\mathbf{x}_0), \mathbf{X}_1')(\mathbf{x}_1' \mid \mathbf{x}_0)\right] \\
&\geq \frac{1}{2}\mathbb{E}[D(\mathbf{X}_1, \mathbf{X}_1')],
\end{aligned}
$$

which finishes the proof. $\qquad\square$

Using the above lemma, to prove Proposition 6, we only need to estimate the expected distance between two batches of samples from $\mu_1$.

*Proof of Proposition 6.* Let $\mathbf{X}_1, \mathbf{X}_1'$ be independent batches of $n$ i.i.d. samples from $\mu_1$. We use expand our notation by letting $D(\mathbf{X}_1, \mathbf{x}_1') \coloneqq \min_{\mathbf{x}_1 \in \mathbf{X}_1} \|\mathbf{x}_1 - \mathbf{x}_1'\|$ be the distance between a single sample and a batch. By the Markov inequality, for any $t > 0$ we have

$$
\begin{aligned}
\mathbb{E}[D(\mathbf{X}_1, \mathbf{X}_1')] &\geq t\mathbb{P}[D(\mathbf{X}_1, \mathbf{X}_1') \geq t] \\
&= t\mathbb{E}\Big[\mathbb{P}\Big[\bigcap_{X_1' \in \mathbf{X}_1'} \{D(\mathbf{X}_1, X_1') \geq t\} \mid \mathbf{X}_1\Big]\Big] \\
&= t\mathbb{E}\Big[\mathbb{P}[D(\mathbf{X}_1, X_1') \geq t \mid \mathbf{X}_1]^n\Big] & \text{(Independence)} \\
&\geq t\mathbb{P}[D(\mathbf{X}_1, X_1') \geq t]^n & \text{(Jensen's Inequality)} \\
&= t\mathbb{E}\Big[\mathbb{P}\Big[D(\mathbf{X}_1, X_1') \geq t \mid X_1'\Big]\Big]^n \\
&= t\mathbb{E}\Big[\mathbb{P}\Big[\bigcap_{X_1 \in \mathbf{X}_1} \{\|X_1 - X_1'\| \geq t\} \mid X_1'\Big]\Big]^n \\
&= t\mathbb{E}\Big[\mathbb{P}[\|X_1 - X_1'\| \geq t \mid X_1']^n\Big]^n \\
&\geq t\mathbb{P}[\|X_1 - X_1'\| \geq t]^{n^2} & \text{(Jensen's Inequality)} \\
&\geq t(1 - Ct^r)^{n^2}. & \text{(Assumption 5)}
\end{aligned}
$$

Choosing $t = (2Cn^2)^{-1/r}$ and using the inequality $(1 - 1/(2x))^x \geq 1/2$ for all $x \geq 1$ yields $\mathbb{E}[D(\mathbf{X}_1, \mathbf{X}_1')] \geq (2Cn^2)^{-1/r}/2$, which completes the proof. $\qquad\square$

## A.2 Using the negative dot-product cost rather than squared-Euclidean in Sinkhorn

As we mention in the main text, entropically regularized optimal transport plan for the squared Euclidean cost can be equivalently recast using exclusively the negative scalar product $(\mathbf{x}, \mathbf{y}) \mapsto -\langle \mathbf{x}, \mathbf{y} \rangle$ between source and target, and not on any absolute measure of scale. To see this, consider an affine map $\overline{\mathbf{x}} = \alpha\mathbf{x} + \beta$ with $\alpha > 0$. Then:

$$
\langle \overline{\mathbf{x}}, \mathbf{y} \rangle = \alpha\langle \mathbf{x}, \mathbf{y} \rangle + \langle \beta, \mathbf{y} \rangle.
$$

The second term is a rank-1 term that will be absorbed by the optimal dual potentials (see (3)) and the factor $\alpha$ amounts to a rescaling of the entropic regularization level $\varepsilon$. In particular, when there is only translation, then $\alpha = 1$ and the transport plans are identical for the same $\varepsilon$.

| | Sinkhorn time, warmstart (s) | | | | Sinkhorn time, no warmstart (s) | | | |
|---|---|---|---|---|---|---|---|---|
| Batch size → | 16384 | 65536 | 262144 | 524288 | 16384 | 65536 | 262144 | 524288 |
| $\varepsilon\downarrow$ | | | | | | | | |
| 0.003 | 3.71 | 133.54 | 1271.23 | 4916.29 | 5.97 | 223.93 | 2300.03 | 9207.89 |
| 0.01 | 1.40 | 48.68 | 466.47 | 1791.55 | 2.02 | 73.64 | 710.40 | 2893.43 |
| 0.03 | 0.49 | 16.09 | 153.78 | 600.50 | 0.65 | 22.01 | 218.63 | 836.82 |
| 0.1 | 0.14 | 3.16 | 31.75 | 126.10 | 0.18 | 5.80 | 61.25 | 229.85 |
| 0.3 | 0.06 | 1.72 | 17.60 | 67.50 | 0.09 | 2.37 | 18.32 | 66.73 |

Table 2: Average per-batch Sinkhorn time in seconds, with and without warmstarting for 32x32 ImageNet OTFM training.

Therefore for input data $\mathbf{X} \in \mathbb{R}^{n\times d}, \mathbf{Y} \in \mathbb{R}^{m\times d}$ the Sinkhorn transport plan depends only on the dot-product cost $-\mathbf{X}\mathbf{Y}^T$. We argue that it is always more natural to use the dot-product cost than the full squared-Euclidean cost, and we find that in practice using directly the dot-product can improve the numerical conditioning of the Sinkhorn algorithm. This is because we drop terms arising from the squared norm, which can be very large. This becomes especially important for single-precision floating point computations, as is the case for the large scale GPU applications we consider.

We illustrate this in Figure 7: we sample $N = 8192$ points $\{\mathbf{x}_i\}_{i=1}^N$ in dimension $d = 128$ from the Gaussian example described in Section 4.2 and map them through the piecewise affine Brenier map, i.e. $\mathbf{y}_i = T(\mathbf{x}_i)$. We then introduce a translation, $\overline{\mathbf{y}}_i = \mathbf{y} + 5$. We use the Sinkhorn algorithm (Algorithm 1) with either the dot-product or squared Euclidean cost to compute the transport plan and record the number of iterations taken, and our computations are carried out on GPU with single-precision arithmetic. Even though we already use log-domain computation tricks to prevent under/overflow, we find that for small $\varepsilon$, Sinkhorn with squared Euclidean cost begins to suffer from numerical issues and fails to converge within the iteration limit of 50,000.

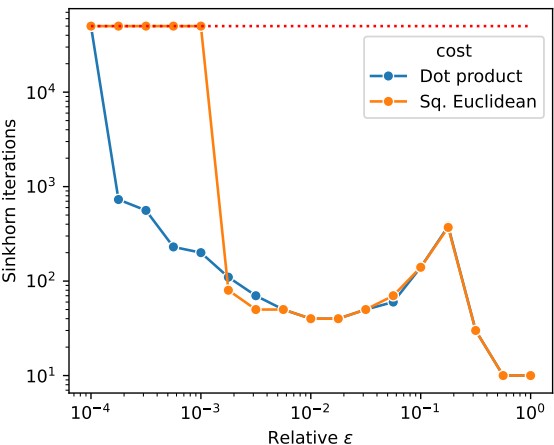

Figure 7: Number of Sinkhorn iterations against relative $\varepsilon$ for Gaussian piecewise affine OT example.

### A.3 Sinkhorn Speedup from Warmstart

Table 2 shows the average time in seconds spent solving (3) using Sinkhorn iterations, for the values of regularization level $\varepsilon$ and batch size $n$ we consider. We find that for almost all choices of $(n, \varepsilon)$, warmstarting yields significant speedups compared to the default Sinkhorn initialization. We therefore enable warmstart by default in our experiments.

### A.4 Sinkhorn Speedup from PCA

Table 3 presents the average wall-clock time of running Sinkhorn on ImageNet-64 with a batch size of 131072 and $\varepsilon = 0.1$. As can be seen, we can reduce dimension by a factor of almost 25, which reduces time by a factor of 10, while having no significant impact on the quality of generated images measured by FID. Moreover, the normalized entropy demonstrates that the coupling obtained from the reduced-dimensional cost matrix has the same sharpness as the original coupling, which is expected since PCA will mostly preserve the dot product cost, resulting in similar couplings.

|  | $k = 500$ | $k = 1000$ | $k = 3000$ | $k = 12288$ (full dimension) |
|---|---|---|---|---|
| *Sinkhorn time* | *1.45s* | *1.82s* | *4.05s* | *14.1s* |
| FID@NFE=4 | 48.4 | 48.1 | 47.0 | 47.3 |
| FID@NFE=8 | 24.7 | 24.4 | 24.0 | 24.2 |
| FID@NFE=16 | 16.0 | 15.8 | 15.8 | 15.8 |
| FID@Dopri5 (Adaptive) | 9.17 | 9.33 | 9.46 | 9.51 |
| Renormalized Entropy | 0.247 | 0.239 | 0.232 | 0.236 |

Table 3: Sinkhorn runtime per batch and FID for different solvers and different PCA dimension $k$. The model is trained on ImageNet-64 with OT batch size = 131072 and $\varepsilon = 0.1$. Note that the difference in FID for full $k$ compared to Table 1 is due to using a different random seed for training.

### A.5 Gaussian Transported with a Piecewise Affine Ground-Truth OT Map

We present in Figure 8 examples of our piecewise affine OT map generation, corresponding to results presented more widely in Figures 2 and 9.

### A.6 Korotin et al. Benchmark Examples

The reader may find examples of the Korotin et al. benchmark in their paper, App. A.1, Figure 6.

### A.7 CIFAR-10 Detailed Results

We show generated images in Figure 12. We see general quantitative and qualitative improvements for larger OT batch size and smaller renormalized entropy. However, these improvements are not as significant as our observation for the more complex down-sampled ImageNet datasets in Appendices A.8 and A.9, likely due to the fact that the dataset size is much smaller. We also plot BPD as a function of renormalized entropy for CIFAR-10, ImageNet-32, and ImageNet-64, in Figure 11.

### A.8 ImageNet-32 Detailed Results

Figure 13 shows generated images using I-FM and OT-FM with different batch sizes and different ODE solvers. As expected, the greatest improvements in the quality of images occur with smaller number of integration steps, which demonstrates the benefit of OT-FM for reducing inference cost.

### A.9 ImageNet-64 Detailed Results

We also perform experiments on the $64 \times 64$ downsampled ImageNet dataset, where we observe an even bigger gap between I-FM and OT-FM with large batch size both in terms of metrics (Figure 6) and in terms of qualitative results (Figure 14). This observation implies that with a proper choice of entropy and batch size, OT-FM is a promising approach to reduce inference cost and generate higher quality high-resolution images.

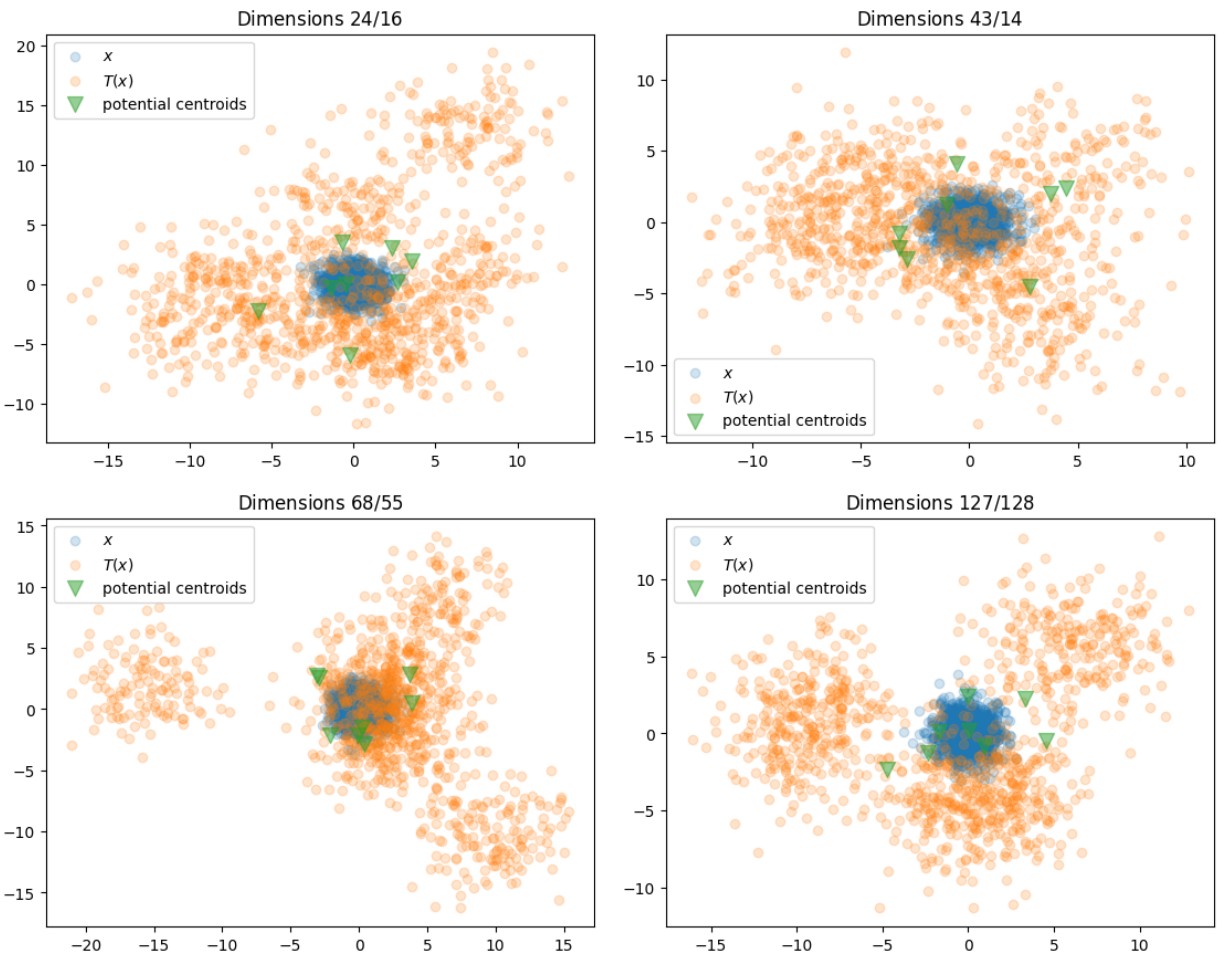

Figure 8: Example of the maps generated in our piecewise affine benchmark task. In these plots $d = 128$ and there are therefore $128/16 = 8$ quadratic potentials sampled around 0. These 2D plots illustrate the action of the same 128 dimensional map, pictured using 2D projections overs pairs chosen in $[1, \ldots, 128]$.

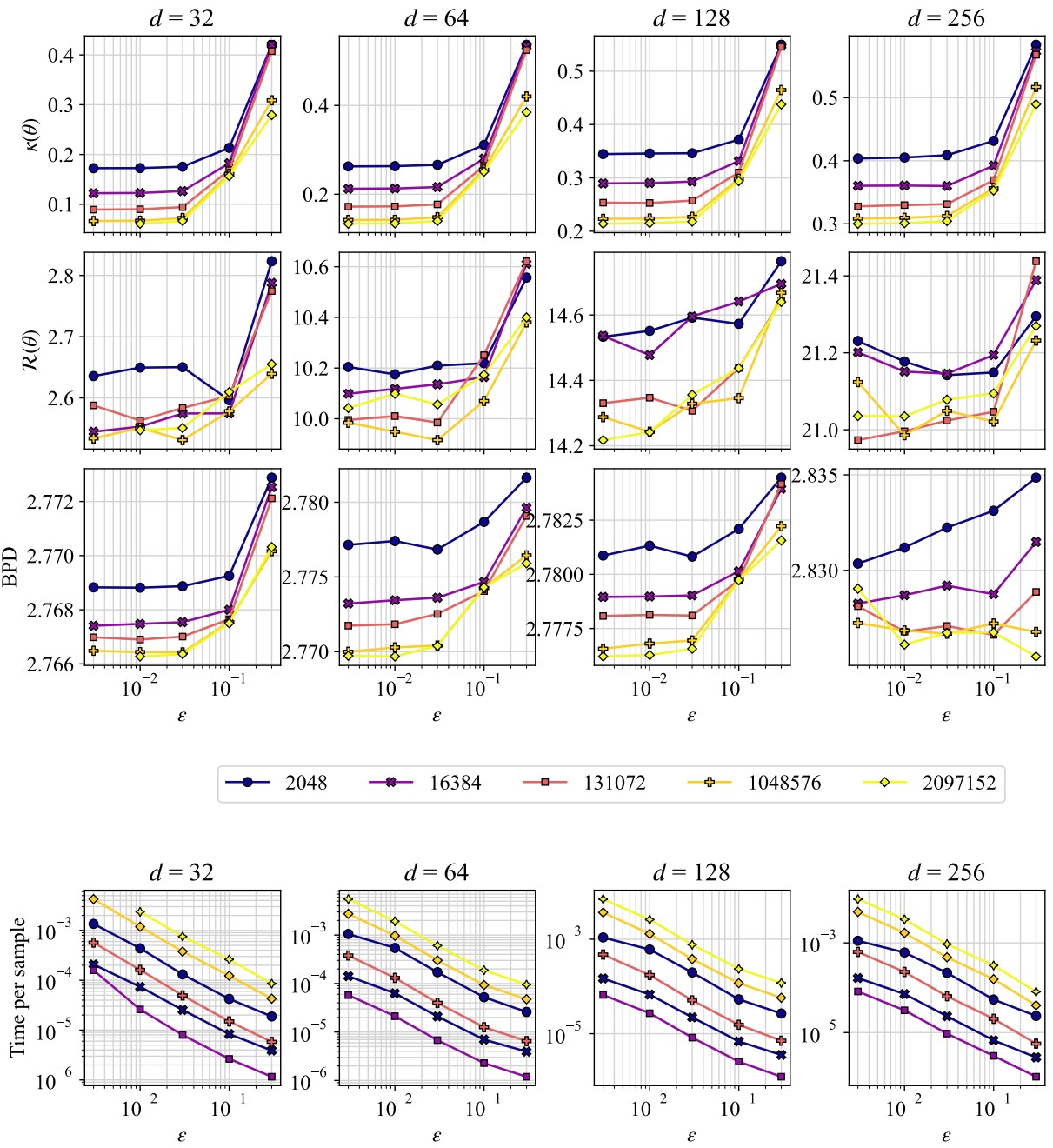

Figure 9: Plots corresponding to Figure 2 in main paper, on piecewise affine synthetic benchmark, using directly the relative `epsilon` parameter as the x-axis (log-scale), instead of re-normalized entropy.

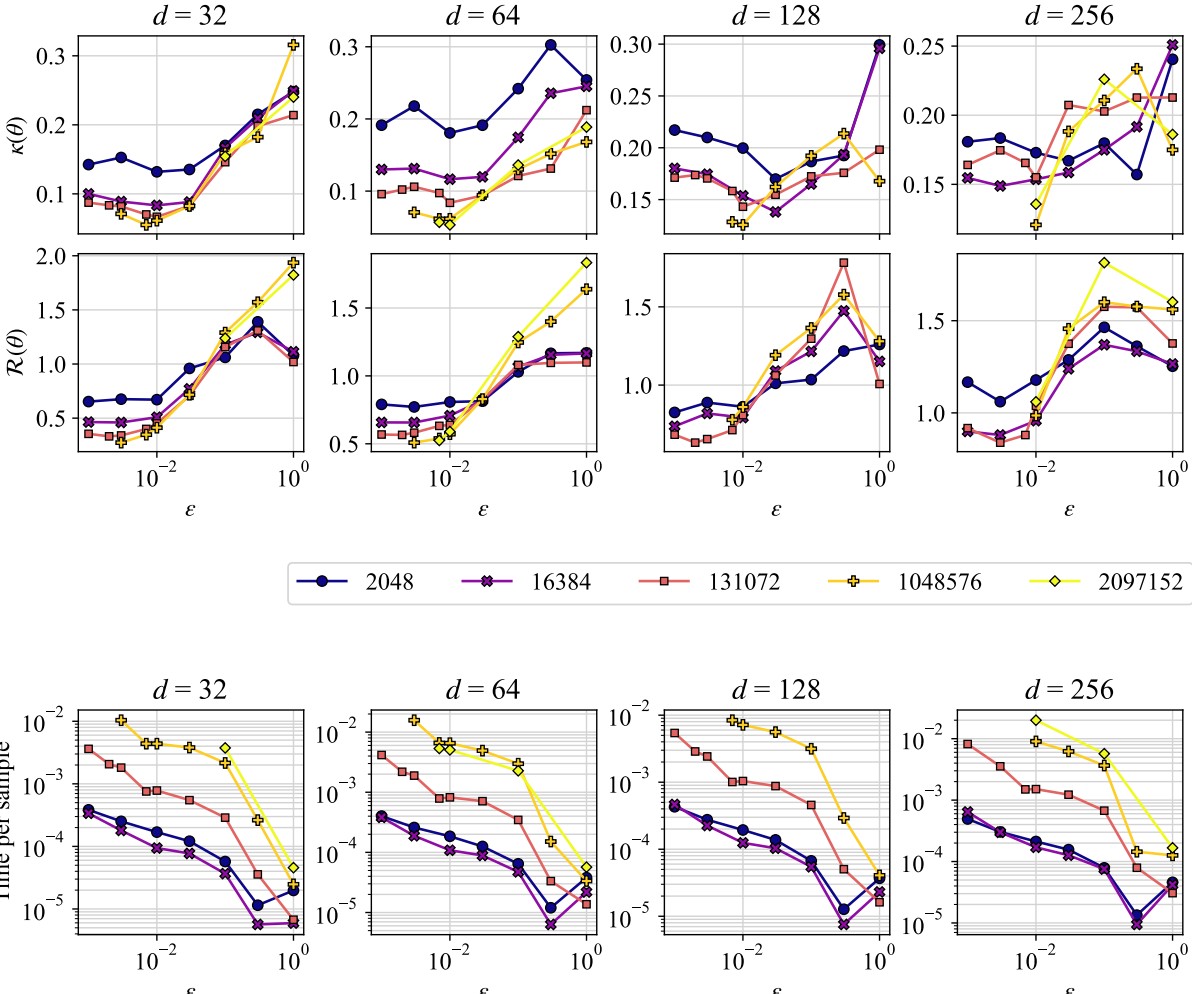

Figure 10: Plots for the Korotin et al. benchmark, shown initially in Figure 3, using the relative `epsilon` $\varepsilon$ parameter directly in the x-axis, in logarithmic scale.

|                          | CIFAR-10        | ImageNet-32       | ImageNet-64 |
| ------------------------ | --------------- | ----------------- | ----------- |
| Channels                 | 256             | 256               | 192         |
| Depth                    | 2               | 3                 | 3           |
| Channels multiple        | 1,2,2,2         | 1,2,2,2           | 1,2,3,4     |
| Heads                    | 4               | 4                 | 4           |
| Heads Channels           | 64              | 64                | 64          |
| Attention resolution     | 16              | 4                 | 8           |
| Dropout                  | 0.0             | 0.0               | 0.1         |
| Batch size               | 128             | 1024              | 800         |
| Iterations               | 400k            | 438k              | 957k        |
| Learning Rate            | 2e-4            | 1e-4              | 1e-4        |
| Learning Rate Scheduler  | Linear-Constant | Polynomial Decay  | Constant    |
| Warmup Steps             | 5k              | 20k               | –           |

Table 4: Hyperparameters used for training flow models for **CIFAR-10** (based on those in Tong et al. (2024)), **ImageNet-32/64** (reproduced from Pooladian et al. (2023)).

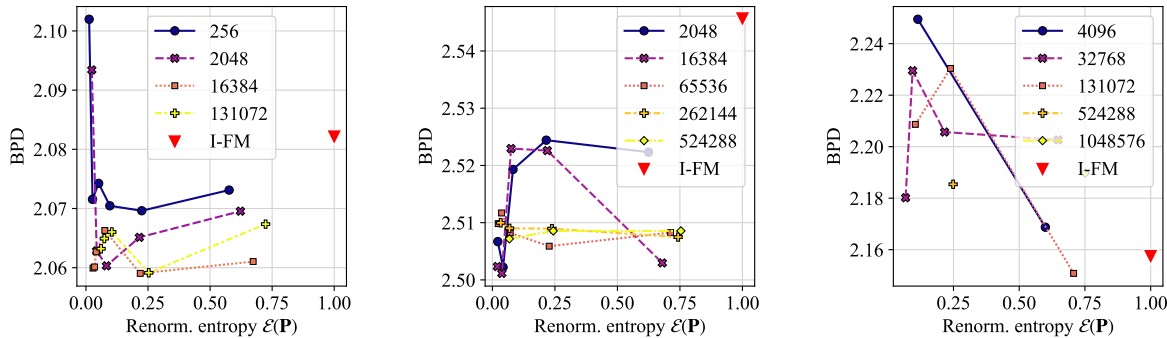

Figure 11: BPD for **CIFAR-10** (Left), **ImageNet-32** (Middle) and **ImageNet-64** (Right). The BPDs are computed using `Dopri5` integration, evaluated on 50 times steps, and computed using 8 vectors for the Hutchinson trace estimator. As a consequence of its high number of function evaluations, the `Dopri5` solver relies less on straightness of the flows. Therefore, we do not observe a significant difference across batch sizes.

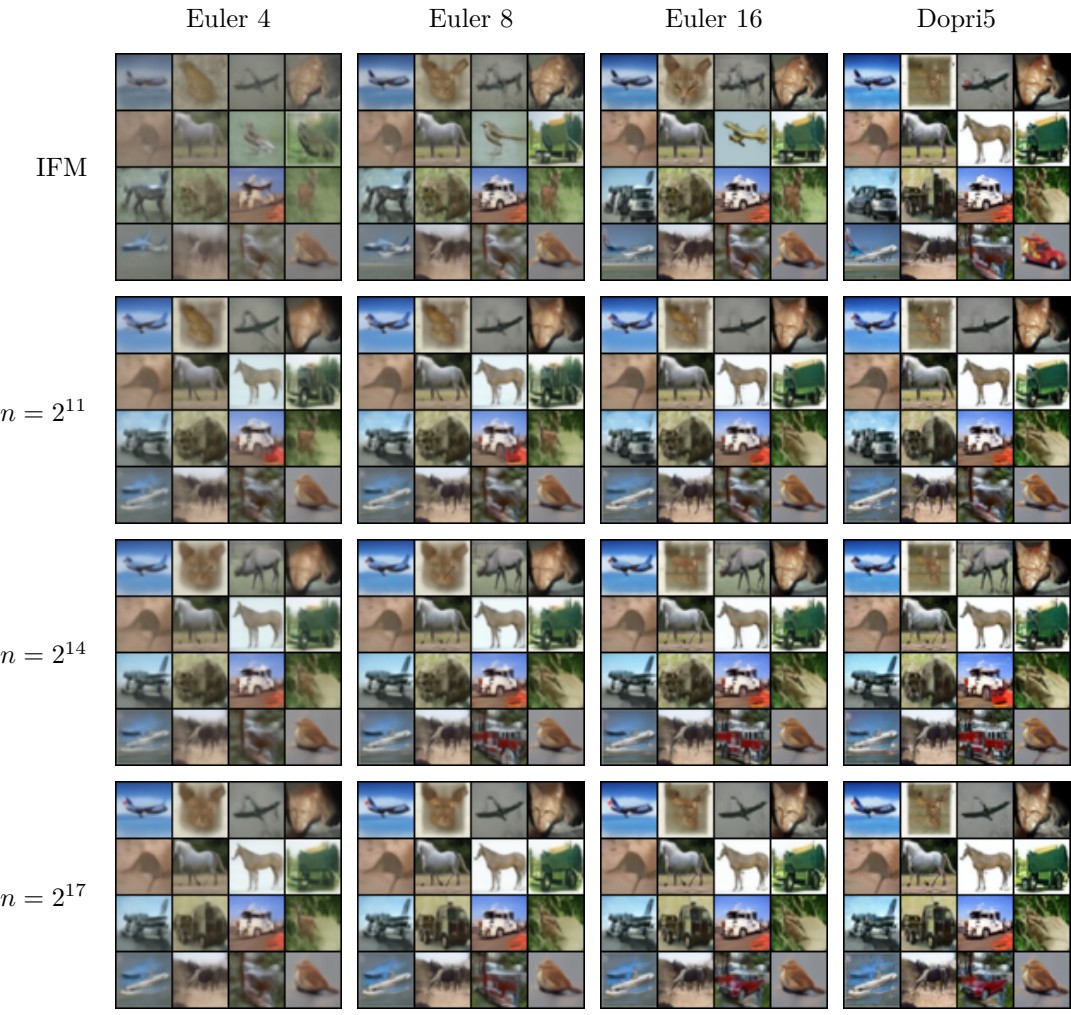

Figure 12: Non-curated images generated from models trained on **CIFAR-10**. The number following `Euler` denotes NFE, while `Dopri5` uses an adaptive number of evaluations. $n$ denotes the total batch size for the Sinkhorn algorithm. We use OT-FM models trained with $\varepsilon = 0.01$.

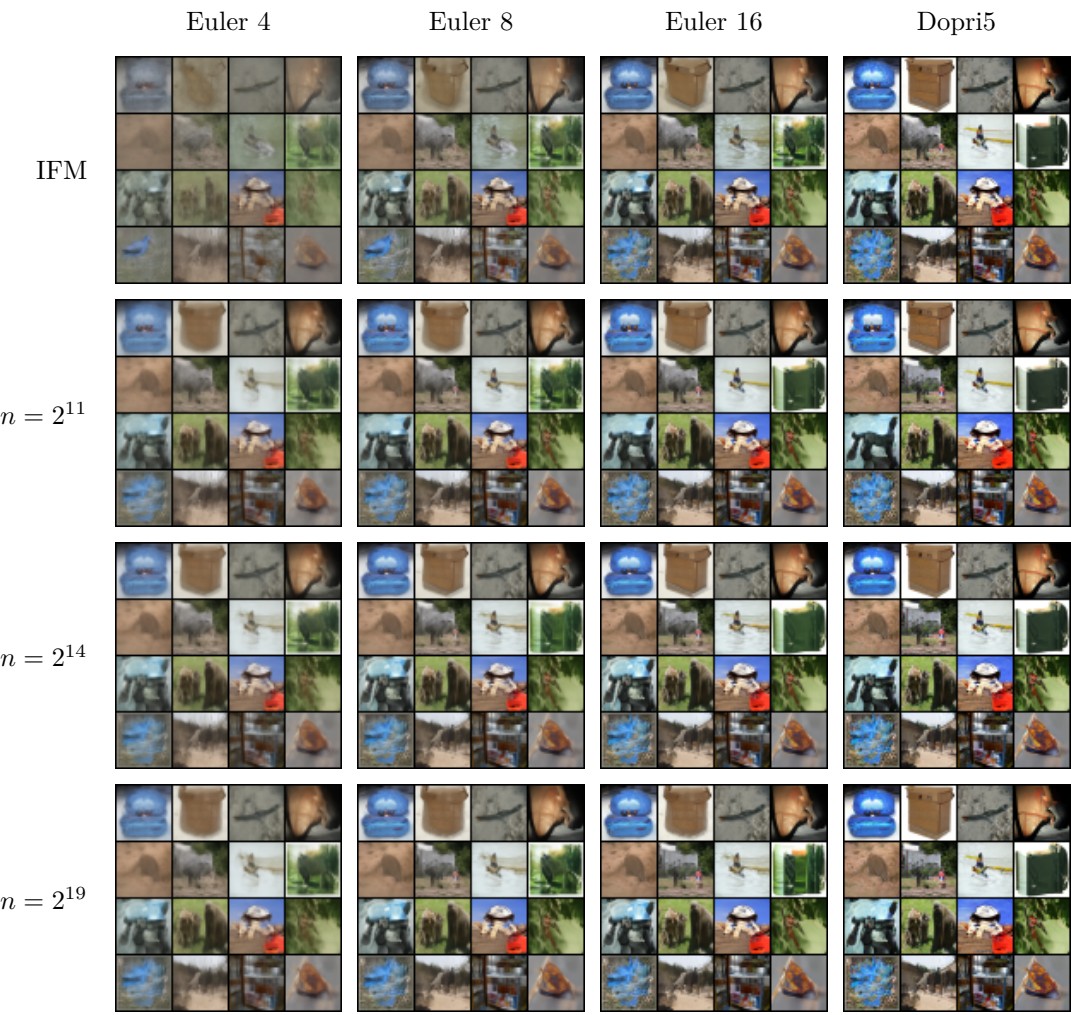

Figure 13: Non-curated images generated from models trained on **ImageNet-32**. $n$ denotes the total batch size for the Sinkhorn algorithm. We use OT-FM models trained with $\varepsilon = 0.1$.

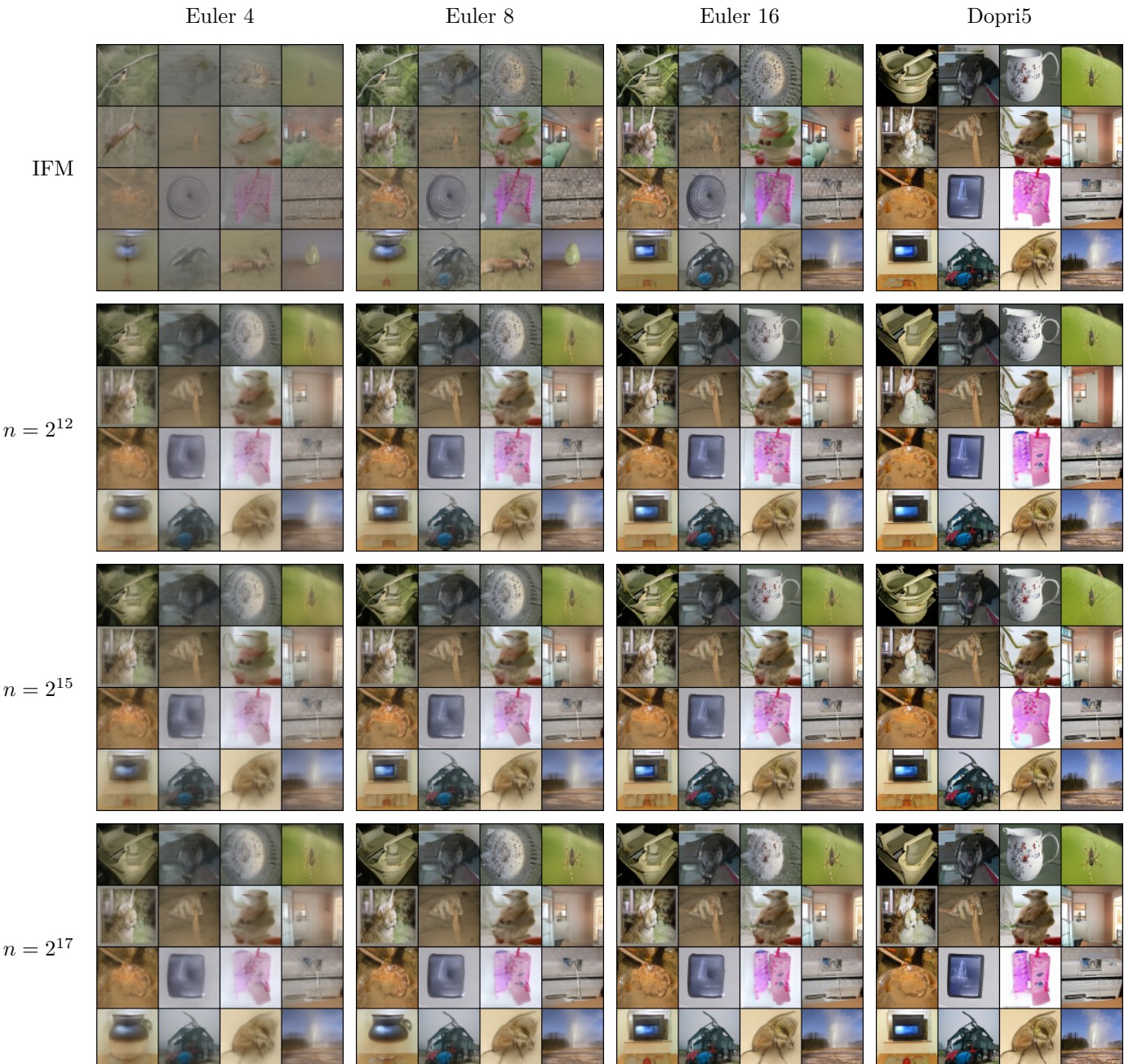

Figure 14: Non-curated images generated from models trained on **ImageNet-64**. $n$ denotes the total batch size for the Sinkhorn algorithm. We use models trained with a varying trained with $\varepsilon = 0.1$.

