# OpenReview forum: "On Fitting Flow Models with Large Sinkhorn Couplings"
_TMLR — Accepted by TMLR_

### Review · Reviewer_UzG3 · 2025-12-04

**Summary Of Contributions:**

This paper aims to study the effect of large batch sizes and varying entropic regularization in the Sinkhorn algorithm when applied to Optimal Transport Flow Matching (OTFM). They provide a theoretical motivation for increasing batch size to achieve more optimal couplings and a renormalized coupling entropy metric for evaluating the amount of blur in the coupling that is independent of batch size and entropic regularization strength. Then, they describe a series of theoretical and engineering adjustments to the Sinkhorn algorithm that enables its performance on very large batch sizes. On low dimensional synthetic tasks and unconditional image generation, they show that generally, OTFM benefits from greater batch sizes and sharper coupling matrices.

Strengths:
- Useful reframing between Hungarian and I-FM in terms of Entropic OT
- Solid theoretical motivation for using large batch size in Batch-OT
- Multitude of well-motivated "tricks" for improving the efficiency of Sinkhorn in large batch settings
- Demonstration of consistent trend for large batch size and low entropic regularization for improved performance in OTFM

Weaknesses
- No ablations on any design choice corresponding to the FM procedure (i.e. learning rate, architecture)
- Lacking (text)-conditioned image generation results, which could be considered the most relevant application of flow matching at the moment

**Audience:**

Yes

**Audience Explanation:**

Yes, flow matching is an active area of research especially in the context of image generation that this paper considers. There is specific interest on scaling up flow matching models as diffusion has been shown to work well at scale, and this work takes steps towards promoting this scalability.

**Claims And Evidence:**

Yes

**Claims Explanation:**

Yes, the claim that OTFM with Sinkhorn benefits from large batch size and low entropic regularization is generally supported across two synthetic benchmarks and two datasets on unconditional image generation.

**Requested Changes:**

- Based on the order of presentation of Section 2, it might be helpful for readers if you define P explicitly as the discrete coupling matrix, especially in regards to equation 3.
- In Figure 6, for the ImageNet-64 results, why is there only one entropy value plotted for the larger batch sizes (500k, 1M)?
- In the definition of renormalized entropy, the range should be from [0, 1] (including 0) in the case that equality holds in the lower bound.
- Curvature definition should sum from 1 to "n", not "N" as it is now
- Is there a reason that text-conditional image generation was not evaluated? I don't believe this is necessary for acceptance because the original flow matching paper does not consider it, but I believe it may strengthen any claims on the applicability of your method to practical applications.

---

> ### Author Response · Authors · 2026-01-10
> **Authors' Response**
>
> We thank the reviewer for their detailed evaluation of our work and are gratified that the reframing of I-FM and batch-OT is well received.
>
> > **Based on the order of presentation of Section 2, it might be helpful for readers if you define P explicitly as the discrete coupling matrix, especially in regards to equation 3.**
>
> We added the definition, thank you for your suggestion.
>
> > **In Figure 6, for the ImageNet-64 results, why is there only one entropy value plotted for the larger batch sizes (500k, 1M)?**
>
> At such a large batch size, computation of Sinkhorn couplings with smaller renormalized entropy (or equivalently, smaller $\varepsilon$) becomes too computationally costly and we could not finish the entire OTFM pipeline for those experiments within 2 weeks. This is due to the $O(1/\varepsilon^2)$ convergence rate of the Sinkhorn algorithm and the $O(n^2)$ computational cost of realigning batches of size $n$.
>
> > **In the definition of renormalized entropy, the range should be from [0, 1] (including 0) in the case that equality holds in the lower bound.**
>
> The Sinkhorn algorithm can only solve the entropic version of the optimal transport problem with $\varepsilon$ strictly positive, and as a result our coupling matrices can never attain exactly the 0 renormalized entropy lower bound.
>
> > **Curvature definition should sum from 1 to "n", not "N" as it is now.**
>
> Thank you for pointing out this typo, we fixed it.
>
> > **Is there a reason that text-conditional image generation was not evaluated? I don't believe this is necessary for acceptance because the original flow matching paper does not consider it, but I believe it may strengthen any claims on the applicability of your method to practical applications.**
>
> We agree that conditional generation is an important application that is compatible with OT flow matching as demonstrated by [1] (among many other works on this topic). For the present paper we restricted our focus to unconditional generation in order to have an “apples-to-apples” comparison to the existing practical works on OTFM for generative modelling, specifically Tong et al. (minibatch-OTFM) and Pooladian et al. (multisample FM).
> We remark that OT couplings for FM can be extended to deal with the case of discrete [1] and continuous [2] conditioning, and we expect the benefits of using large batches for OTFM to carry over to the conditional setup as well.
>
> ---
> References
>
> [1] J. Chemseddine et al. “Conditional Wasserstein Distances with Applications in Bayesian OT Flow Matching.” JMLR 2025.
>
> [2] Kerrigan G, Migliorini G, Smyth P. “Dynamic conditional optimal transport through simulation-free flows.” Advances in Neural Information Processing Systems. 2024 Dec 16;37:93602-42.

---

### Review · Reviewer_mxy7 · 2025-12-04

**Summary Of Contributions:**

The paper presents a study into the importance of finding good data-noise couplings when training ODE-based generative models. The authors focus on scaling batch optimal transport to very large batch sizes in order to find couplings which are closer to the underlying Monge OT map. To do this they introduce various extensions of existing Sinkhorn algorithm implementations, including rescaling Sinkhorn's epsilon parameter depending on the given cost matrix. using dot-products to compute the cost matrix, warm-starting the Sinkhorn optimisation using the previous batch, using PCA to reduce the dimensionality of data required for constructing the cost matrix. The authors also propose a number of adaptions to Sinkhorn computation to allow scaling to very large batch sizes. Their results show that increasing the batch size for batch OT and reducing the epsilon parameter are important for generative performance and can lead to significant gains over an independent noise-data coupling, which is commonly used.

**Audience:**

Yes

**Audience Explanation:**

The paper's results show convincing evidence that scaling batch OT to very large batch sizes can lead to significant improvements in performance over independent noise-data coupling which is very commonly used to train flow matching generative models. Although these improvements do seem to diminish for larger batch sizes, especially when more sampling steps are used or when sampling with adaptive ODE solvers.

**Claims And Evidence:**

Yes

**Claims Explanation:**

The authors provide significant evidence for their claims with experiments on numerous datasets, including both synthetic data where exact optimal couplings are available, as well commonly used Image benchmark datasets. All of their experiments involve comparisons with independent coupling and often show significant performance improvements over this setup. The authors are also up front about the limitations of their approach including the enormous computational cost of computing Sinkhorn couplings for very large batch sizes.

**Requested Changes:**

### Minor Changes

- I think it could be helpful to outline that P is the probability matrix that the couplings will be sampled from in equation (3), since it does not  seem to be introduced before.
- I find the experiments in Appendix 4 convincing, but I think it could be helpful to add to the experiments section of the main text which values of K were used for each dataset.
- The notation for $\boldsymbol{C}$ should be made consistent throughout, sometimes it is shown as $- \boldsymbol{X}^T \boldsymbol{Y}$ and sometimes  $- \boldsymbol{X}\boldsymbol{Y}^T$

### Other Questions

- In Figures 2 and 3 it seems that the cost of computing the coupling is almost always higher for batch size 2048 than batch size 16384. Why is this?
- In Figure 3 why are some of the plots missing results for low renormalised entropy (especially for some plots with higher batch sizes)?

---

> ### Author Response · Authors · 2026-01-10
> **Authors' Response**
>
> We thank the reviewer for their time and effort on our manuscript and for their positive evaluation of our contributions.
>
> > **Minor Changes**
>
> Thank you for your comments, we have incorporated all of them.
>
> > **In Figures 2 and 3 it seems that the cost of computing the coupling is almost always higher for batch size 2048 than batch size 16384. Why is this?**
>
> Since the Sinkhorn computations are sharded, there may be communication/computation tradeoffs that perturb the simpler understanding that large n necessarily results in more time. For this reason, we expect that for moderate batch sizes $n$ the computation time won’t grow significantly as $n$ grows. As a result, one gets a smaller *per-sample time* (which is the total time divided by $n$) for larger $n$ (up to 16,384). The total time however would behave as expected, i.e. increasing monotonically with $n$.
>
> > **In Figure 3 why are some of the plots missing results for low renormalised entropy (especially for some plots with higher batch sizes)?**
>
> These runs did not finish successfully due to significant computational requirements, as the combination of low renormalized entropy (or equivalently low $\varepsilon$) and large batch size induces a huge cost due to the $O(1/\varepsilon^2)$ convergence rate of the Sinkhorn algorithm and the $O(n^2)$ computational cost of each iteration.

---

> > ### Comment · Reviewer_mxy7 · 2026-01-16
> >
> > Thank you for your answers and updates to the paper. I think the revised manuscript is very good and begins to answer a very interesting research question.

---

### Review · Reviewer_cnAE · 2025-12-15

**Summary Of Contributions:**

In this manuscript, the authors focus on the optimization of the Optimal Transport (OT) coupling in the training of flow models. The core lies in enhancing the training efficiency and generation performance of flow models by expanding the batch size of the Sinkhorn algorithm and optimizing the entropy regularization parameter. The manuscript meets TMLR’s standards for novelty, technical rigor, and empirical evaluation, and we recommend acceptance subject to minor revisions to improve clarity and reproducibility.

**Audience:**

Yes

**Audience Explanation:**

First, I believe that individuals in fields such as computer vision and medical imaging will undoubtedly find this interesting. Moreover, it breaks through the large-scale computational bottleneck, offering practical value in engineering applications.

**Broader Impact Concerns:**

The proposed revisions are targeted at resolving minor gaps in reproducibility, clarity, and structural coherence without altering the manuscript’s core contributions. We look forward to reviewing the revised version of the manuscript.

**Claims And Evidence:**

Yes

**Claims Explanation:**

It provides rigorous theoretical proof. The manuscript establishes the theoretical foundation for large-scale Sinkhorn couplings through proposition derivations, the application of Brenier’s theorem, the continuous interpolation properties of entropic regularization, and complexity analysis via piecewise computation.
It includes reproducible experiments with real data, supported by ablation studies.

**Requested Changes:**

Minor revision
1.	Please provide a public repository for the sharded Sinkhorn implementation (built on OTT-JAX) and flow matching training pipelines, with detailed environment setup instructions (e.g., GPU cluster configuration, JAX sharding parameters). This will enable other researchers to replicate large-batch OT-FM experiments.
2.	Please revise informal or imprecise expressions. For instance, in the introduction, replace the colloquial phrase "the jury seems to be still out on ruling" with the more academic "it remains unclear whether".
3.	In Table 2,the column header "Batch size ε" is ambiguous. Please add directional arrows to it, following the same format as used in the header of Table 1.
4.	Please standardize metric abbreviations (e.g., define NFE on first use in Section 4.1, not in Figure captions).
5.	There is no description of Renorm. Entropy (P).
6.	In ‘4.3 Unconditioned Image Generation (Training Setup)’, if you want to describe the random seed strategy and learning rate schedule, please describe it completely.
7.	Please enhance the cohesion between the main text and Appendix A, it is recommended to integrate explicit cross-references (e.g., "Refer to Appendix A.x for further details") at corresponding locations within the main body.
8.	Please revise the reference format to comply with the TMLR submission specifications, ensuring consistency across all entries (e.g., the formatting for conferences like ICLR and ICML is currently inconsistent).

---

> ### Author Response · Authors · 2026-01-10
> **Authors' Response**
>
> We thank the reviewer for their detailed reading of our submission and we are delighted to read that the reviewer agrees our work will be impactful and interesting to e.g. vision and imaging communities. We address point-by-point the questions raised in the following.
>
> > **1. Please provide a public repository for the sharded Sinkhorn implementation (built on OTT-JAX) and flow matching training pipelines, with detailed environment setup instructions (e.g., GPU cluster configuration, JAX sharding parameters).**
>
> We commit to providing a repository containing code and scripts for reproducing the results of the paper upon acceptance.
> Sharded Sinkhorn, which forms the core focus of our article, is *available out of the box* in OTT-JAX through natively supported shardings available in JAX (see p6 of our submission and also e.g. https://ott-jax.readthedocs.io/tutorials/linear/900_CIFAR_benchmark.html for an illustrative example of how this can be done).
>
> We will prepare supporting scripts for loading and running experiments in addition to these. For transparency we also answer here directly regarding:
> *JAX sharding parameters*: For sharded Sinkhorn between $(x_0, x_1)$, both input arrays are sharded along the data dimension $N$ across available devices.
> *GPU cluster details*: Minimal custom configuration is required as we rely on JAX parallelization. Our experiments were run on 1x8 or 2x8 A100 or H100 nodes (see also details in p6 of our submission).
>
> > **2. Please revise informal or imprecise expressions ...**
> > **3. In Table 2, the column header "Batch size $\varepsilon$" is ambiguous ...**
> > **4. Please standardize metric abbreviations ...**
>
> Thank you for your suggestions. We have incorporated them in the revised version.
>
> > **5. There is no description of Renorm. Entropy (P).**
>
> We explained renormalized entropy in a paragraph at the top of page 5. We’d be happy to provide further clarification if you believe it would be helpful.
>
> > **6. In ‘4.3 Unconditioned Image Generation (Training Setup)’, if you want to describe the random seed strategy and learning rate schedule, please describe it completely.**
>
> Thank you for pointing this out, we agree that hyperparameter details are important to clarify and have added explicit references to the relevant section and table in the cited papers.  We have provided additional details in the revision (Table 4).
>
> Initially we had omitted discussion of the learning rate schedule and other implementation details for brevity, since we reproduced exactly the training strategy employed in the methods we compare to, instead providing the references that we follow for our implementation.
>
> > **7. Please enhance the cohesion between the main text and Appendix A, it is recommended to integrate explicit cross-references (e.g., "Refer to Appendix A.x for further details") at corresponding locations within the main body.**
>
> Thank you for pointing this out, we had originally neglected to cross-reference several subsections (A.6-A.9). This has been rectified.
>
> > **8. Please revise the reference format to comply with the TMLR submission specifications, ensuring consistency across all entries (e.g., the formatting for conferences like ICLR and ICML is currently inconsistent).**
>
> Thank you for pointing this out. We follow the author-year citation style of TMLR, and use the `\citeauthor` macro of `natbib` to highlight famous names/results to facilitate reading. We did find a number of citations that were part of the text and we forgot to remove parentheses, which were fixed.

---

> > ### Comment · Reviewer_cnAE · 2026-01-14
> > **Thank you to the authors for your thoughtful and detailed response to the revision comments!**
> >
> > I have reviewed all the revision requests I previously raised, and confirm that all issues have been properly addressed. The revised content is clear and thorough, which further improves the reproducibility and readability of the paper.
> > The paper now meets the acceptance criteria, and I recommend proceeding with the subsequent processes.

---

> > > ### Comment · Action_Editor_5BYV · 2026-01-31
> > > **Recommendation**
> > >
> > > Dear Reviewer,
> > >
> > > Thanks for your review and discussion. Could you please submit the recommendation?
> > >
> > > Best reegards,
> > >
> > > The AE

---

### Decision · Action_Editor_5BYV · 2026-03-27

**Recommendation:** Accept as is

**Audience:**

Yes

**Audience Explanation:**

This manuscript presented solid engineering-oriented research. The authors investigated an interesting research question: how batch size influences performance in the context of batch optimal transport. They proposed several new techniques to enable the use of extremely large batch sizes in practice. The proposed methods and empirical findings are expected to attract significant interest from researchers working on generative models TMLR.

**Claims And Evidence:**

Yes

**Claims Explanation:**

This work provides a theoretical basis for large-scale Sinkhorn couplings through proposition derivations, Brenier’s theorem, entropic regularization analysis, and complexity analysis, along with reproducible real-data experiments and ablations. It also theoretically and empirically shows that large batch sizes and weak entropic regularization enhance OTFM–Sinkhorn performance in synthetic tasks and unconditional image generation, offering valuable scaling guidance for the active flow-matching research community, particularly for image generation. All the reviewers consistently acknowledged the claims supported with accurate, convincing and clear evidence.